# Loss of MTX2 causes mandibuloacral dysplasia and links mitochondrial dysfunction to altered nuclear morphology

Sahar Elouej et al.[#]

Mandibuloacral dysplasia syndromes are mainly due to recessive *LMNA* or *ZMPSTE24* mutations, with cardinal nuclear morphological abnormalities and dysfunction. We report five homozygous null mutations in *MTX2*, encoding Metaxin-2 (MTX2), an outer mitochondrial membrane protein, in patients presenting with a severe laminopathy-like mandibuloacral dysplasia characterized by growth retardation, bone resorption, arterial calcification, renal glomerulosclerosis and severe hypertension. Loss of MTX2 in patients' primary fibroblasts leads to loss of Metaxin-1 (MTX1) and mitochondrial dysfunction, including network fragmentation and oxidative phosphorylation impairment. Furthermore, patients' fibroblasts are resistant to induced apoptosis, leading to increased cell senescence and mitophagy and reduced proliferation. Interestingly, secondary nuclear morphological defects are observed in both *MTX2*-mutant fibroblasts and mtx-2-depleted *C. elegans*. We thus report the identification of a severe premature aging syndrome revealing an unsuspected link between mitochondrial composition and function and nuclear morphology, establishing a pathophysiological link with premature aging laminopathies and likely explaining common clinical features.

[#]A list of authors and their affiliations appears at the end of the paper.

Mandibuloacral dysplasia (MAD) is a rare, mostly autosomal recessive, progeroid disorder with clinical and genetic heterogeneity, characterized by growth retardation, craniofacial dysmorphic features due to distal bone resorption, musculoskeletal and skin abnormalities associated with lipodystrophy. Two major types of MAD are differentiated according to body fat distribution patterns and genetic origins: type A (MADA) characterized by partial lipodystrophy, caused by mutations of the Lamin A/C (*LMNA*) gene[1] and type B (MADB) with generalized lipodystrophy, caused by mutations of the zinc metalloproteinase (*ZMPSTE24*) gene[2]. MAD is also a feature of Nestor–Guillermo Progeria Syndrome, caused by recessive *BANF1* mutations[3], and of Mandibular hypoplasia, Deafness, Progeroid features, and Lipodystrophy (MDPL) syndrome[4–6], another rare disorder due to dominant mutations in the *POLD1* gene, encoding the catalytic subunit of DNA Polymerase Delta 1. MADA linked to *LMNA* recessive mutations was the first MAD syndrome with an identified gene, and also the first systemic laminopathy to be described in 2002, while previously described laminopathies only affected selected tissues[7–9]. Due to overt clinical similarities with MADA, our group searched for *LMNA* mutations in patients affected with Hutchinson–Gilford Progeria Syndrome (HGPS)[10] and identified a recurrent de novo *LMNA* mutation affecting Lamin A splicing and causing the accumulation of a toxic prelamin A derivative called Progerin[11,12]. Notably, MADB[13], MADA, and several other laminopathies with lipodystrophy[14–16] are also characterized by toxic Prelamin A accumulation, suggesting a molecular link to the observed phenotypic continuum. Since their discovery, the diverse molecular dysfunctions linked to nuclear accumulation of prelamin A, and namely its derivative progerin in HGPS, together with possible therapeutic options, have been extensively studied[17–19]. Among these dysfunctions, typical nuclear morphology defects as well as altered mitochondrial function, with altered respiratory chain composition and reduced ATP production, have been documented in HGPS[20,21]. Here, we report a novel MAD progeroid syndrome (MADaM: Mandibuloacral dysplasia associated to *MTX2*) with clinical features resembling HGPS[22,23], due to recessive mutations in *MTX2* encoding Metaxin-2 (MTX2), an outer mitochondrial membrane (OMM) protein.

The functions of MTX2, a 263-aa ubiquitously expressed protein (Uniprot O75431), remain largely unknown. MTX2 is located at the OMM and faces the cytosolic compartment through direct interaction with its partner Metaxin-1 (MTX1)[24,25]. Both proteins are involved in protein translocation into mitochondria[24], being part of the mitochondrial sorting and assembly machinery (SAM) responsible for the correct integration of β-barrel proteins into the OMM[25–30].

The SAM complex is a binding partner of Mic60, a MICOS component involved in cristae junction (CJ) organization and maintenance[27]. These interactions form large "mitochondrial intermembrane space bridging" complexes that help shaping and stabilizing CJs[31,32]. In addition, several studies identified a role for MTX1 and MTX2 in TNF-α-induced apoptosis, through the interaction with the pro-apoptotic protein Bak[33–36]. The functional studies we report in MADaM patient-derived primary fibroblasts show that the loss of MTX2 and MTX1 causes mitochondrial network fragmentation, decreased oxidative phosphorylation, resistance to apoptosis triggering by TNF-α, increased senescence and autophagy, and reduced proliferation; furthermore, it secondarily impacts nuclear morphology in a fashion that resembles HGPS and other progeroid laminopathies, probably underlying common clinical features.

## Results

### Genotypes and clinical findings in MADaM patients.

We evaluated seven patients issued from five consanguineous unions, originating from India (MADM1), Turkey (MADM2), Algeria (MADM3), Egypt (MADM4-1 to 3), and Ecuador (MADM5), diagnosed with severe progeroid form of MAD with growth retardation, small viscerocranium with mandibular underdevelopment, distal acro-osteolyses, lipodystrophy, altered skin pigmentation, renal focal glomerulosclerosis, and extremely severe hypertension in most cases, eventually associated with disseminated arterial calcification (MADM3) (Fig. 1, Supplementary Note 1, Supplementary Data File, and Supplementary Figs. 1, 2, 3a–e for further clinical, genealogical and radiological observations). All patients had normal cognitive development. After inconclusive molecular diagnosis for the *LMNA, ZMPSTE24, BANF1,* and *POLD1* genes, whole-genome sequencing (WGS) was performed on the first three index cases and the parents of MADM1 and 3. Due to the strong similarities in clinical presentation and progression, these patients were expected to be affected by the same disorder. Based on the consanguinity of healthy parents of each patient, the bioinformatic analysis of WGS data was oriented toward identification of homozygous variants in a common gene (autosomal recessive inheritance with identity-by-descent)[37]. This analysis indeed identified three different homozygous sequence variants in the *MTX2* gene (NM_006554.4) encoding MTX2: c.2T>A p.(Met1-Lys) in patient MADM1 (exon 1), c.544-1G>C (within the splice acceptor site in intron 8) in patient MADM2 and c.208+3_208+6del (within the splice donor site in intron 4) in patient MADM3. The parents of MADM1, MADM3, and MADM4-3 were heterozygous, consistent with autosomal recessive transmission (Fig. 1a, c, d). Patients MADM4-1 and 2, whose DNA was available from the 4th family, were exome-sequenced independently and found to carry a germline homozygous *MTX2* frameshifting mutation in exon 9: c.603del, p.(Tyr202Ilefs*26) (Fig. 1). In addition, independent exome sequencing of patient MADM5 revealed a homozygous 2-bp deletion in exon 6 of *MTX2*, c.294_295delTC, resulting in a direct nonsense mutation p.(Leu99*). Although consanguinity was not known for the parents of the patient, they were born in the same region from Ecuador and the search for regions of homozygosity (ROH) in the patient's exome data (Supplementary Fig. 3f), showed that altogether autosomal ROH spanned about 34 Mb (1/88 of the autosomal genome), arguing for ancient inbreeding[38]. Moreover, the *MTX2* mutation was located within the largest autosomal ROH, spanning 5 Mb. None of the identified mutations was reported in dbSNP144, 1000 Genomes Project, GnomAD (including ExAC), nor Exome Variant Server databases and all were predicted as deleterious according to UMD predictor, SIFT, Mutation-Taster prediction tools; Varsome classed all the variants as being "variants of unknown significance". p.Met1 was phylogenetically conserved (Supplementary Fig. 4) and the c.2T>A variant prevented translation initiation; the two splice variants were predicted to impact splicing by Human Splicing Finder (data not shown), and the two frameshifting deletions caused a premature termination codon: all mutations thus led most probably to the absence of MTX2.

These five *MTX2* mutations are the first reported in humans.

### Functional studies in fibroblast cell lines from patients.

To confirm the pathogenicity of these *MTX2* mutations in the context of MADaM syndrome, functional in vitro studies were performed on cutaneous primary fibroblasts from patients MADM2 and MADM3. cDNA sequence analysis showed a deletion of 7 bp at the beginning of exon 9 for patient MADM2 (c.544_550del)

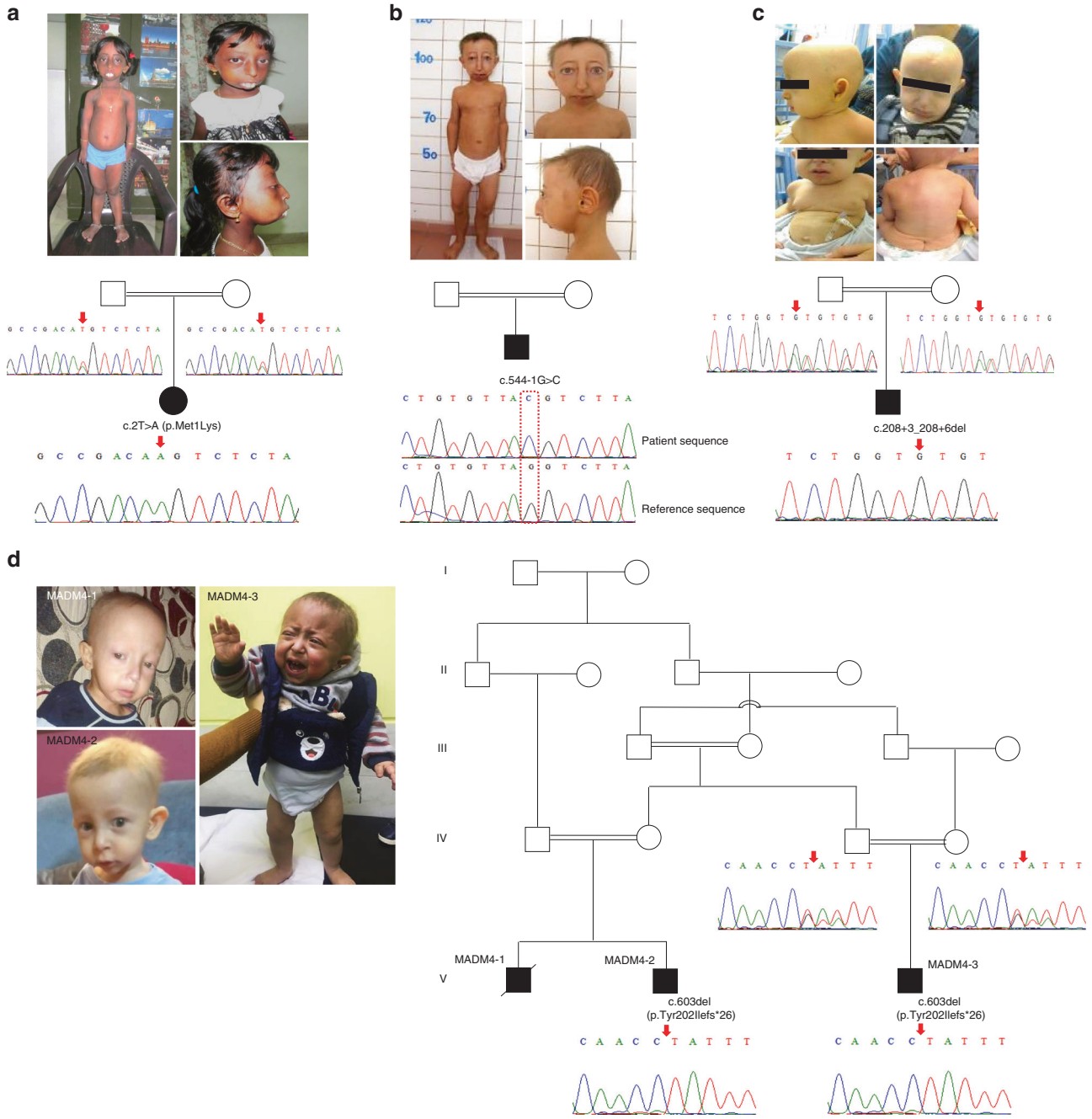

**Fig. 1 Patients' clinical features and *MTX2* mutations. a** Pictures and electropherograms of patient MADM1. At age 11 years, the girl showed severe growth retardation, mandibular recession, a pinched nose with hypoplastic *alae nasi*, long philtrum, prominent eyes, dental overcrowding, sparse eyebrows and body hair along with lipodystrophy and atrophic wrinkled skin. The homozygous c.2T>A (p.Met1Lys) *MTX2* variant was identified and the heterozygous state of the parents confirmed the mode of inheritance as autosomal recessive. **b** Pictures and electropherograms of patient MADM2 at age 14 years, showing clinical features very similar to patient MADM1. The homozygous c.544-1G>C *MTX2* splicing variant was confirmed by Sanger sequencing. **c** Pictures and electropherograms of patient MADM3 at age 2 years. He showed mandibular recession, prominent eyes, alopecia, long Stahl's ears, and lipodystrophy. The homozygous c.208+3_208+6del *MTX2* splicing variant was observed and confirmed by Sanger sequencing, and the carrier state of the parents was in accordance with autosomal recessive inheritance pattern. **d** Pictures of patients and family tree of the MADM4 consanguineous family. The homozygous *MTX2* deletion c.603del (p.Tyr202Ilefs*26) was identified in three affected children (MADM4-1, MADM4-2, MADM4-3), all presenting with progeroid facies including high forehead, bulbous nose with depressed nasal bridge, prominent eyes, long philtrum, small mouth, small mandible, and long Stahl's ears. Informed consent was obtained to publish patient images. See Supplementary Note 1, Supplementary Figs. 1, 2, and Supplementary Data File for additional clinical and radiological features.

and deletion of the entire exon 4 for patient MADM3 (c.136_208del), both leading to premature termination codons (respectively p.Val182Argfs*3 and p.Ala46Valfs*12) (Supplementary Fig. 4). We next examined MTX2 expression and the

abundance of other mitochondrial membrane proteins in control and patients' fibroblasts. Consistent with the possible instability of the truncated MTX2 proteins, we failed to detect the corresponding ~30 kDa band corresponding to endogenous MTX2 by

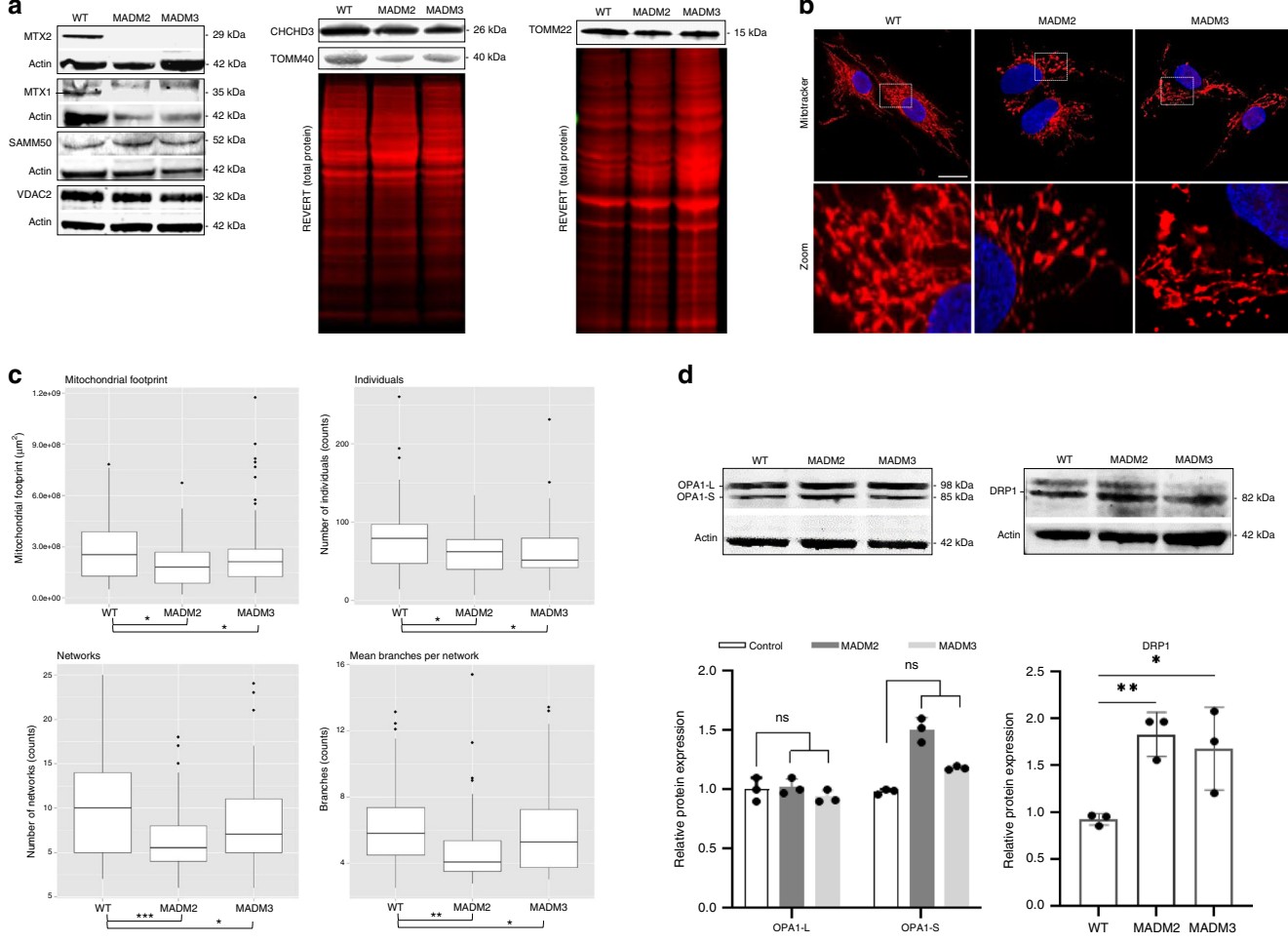

**Fig. 2 MTX2 deficiency induces altered mitochondrial protein composition and increased mitochondrial fission. a** Immunoblot analysis of MTX2 and other outer and inner mitochondrial membrane proteins of whole-cell extracts from healthy control (WT) and patients' (MADM2, MADM3) fibroblasts. Actin or REVERT total proteins were used as loading controls. $n = 3$ independent experiments for MTX2, TOMM22, SAMM50, CHCHD3 and $n = 2$ independent experiments for MTX1, VDAC2, TOMM40. **b** Representative confocal microscopy images showing mitochondria stained using DAPI (blue) and Mitotracker (red) from a control, patient MADM2, and patient MADM3. Boxed regions are enlarged. Scale bar: 10 μm. Data shown are representative of ten independent experiments. **c** Mitochondrial network analysis using the MINA ImageJ macro of healthy and patients' fibroblasts showing significant differences in: mitochondrial footprint (exact $p$ values: $p_{MADM2} = 0.026$, $p_{MADM3} = 0.041$), individuals ($p_{MADM2} = 0.013$, $p_{MADM3} = 0.042$), networks ($p_{MADM2} = 7.31737E{-}05$, $p_{MADM3} = 0.041$) and mean number of branches per network ($p_{MADM2} = 0.005$, $p_{MADM3} = 0.043$). 53 (WT), 46 (MADM2) and 51 (MADM3) cells from $n = 5$ independent experiments were blindly scored. Box plots show median (horizontal lines), first to third quartile (box), and the most extreme values within 1.5 times the interquartile range (vertical lines). Two-tailed unpaired $t$ test; *$p < 0.05$, **$p < 0.01$, ***$p < 0.001$. **d** Immunoblot analysis of OPA1-L (protein optic atrophy 1, long) and OPA1-S (protein optic atrophy 1, short) (proteins involved in mitochondrial fusion, OPA1-S being issued from cleavage of OPA1-L) and DRP1 (protein involved in mitochondrial fission). Protein levels were quantified by ImageJ software and their expression levels were normalized to actin as an internal loading control. Results are expressed as mean ± SD, two-tailed unpaired $t$ test was used to evaluate the statistical significance of differences among the groups (exact $p$ values: OPA1-L: $p_{MADM2} = 0.772$, $p_{MADM3} = 0.385$; OPA1-S: $p_{MADM2} = 0.10$, $p_{MADM3} = 0.10$; DRP1: $p_{MADM2} = 0.0015$, $p_{MADM3} = 0.042$). *$p < 0.05$, **$p < 0.01$, ns not significant ($n = 3$ and $n = 4$ independent experiments respectively for OPA1 and DRP1). Source data are provided as a Source Data file.

western blot (Fig. 2a). While *MTX1* genomic coding sequences were wild type in patients MADM2 and 3 (data not shown), we observed a complete secondary depletion of MTX1 in their fibroblast cell lines (Fig. 2a). These results were confirmed in fibroblasts from patients MADM4-2 and MADM4-3 (Supplementary Fig. 5). This depletion was not due to a reduction of *MTX1* transcripts' levels, as verified on fibroblasts of patients MADM2 and 3, (Supplementary Fig. 6), and correlates with an earlier study reporting that MTX2 depletion leads to a similar reduction of MTX1 protein levels[31].

Conversely, the levels of other outer and inner mitochondrial membrane proteins, tested by western blot, were unaffected (Fig. 2a).

**Mitochondrial network alterations in MADaM patients**. To study the morphology and structure of the mitochondrial network, mitochondria were stained by MitoTracker Red. Patients' fibroblasts displayed network fragmentation with striking mitochondrial aggregates, compared to control cells (Fig. 2b). ImageJ (MiNA macro) analyses evidenced reduced mitochondrial footprints with significantly fewer mitochondria ("individuals"), and fewer branches per network in MTX2-deficient cells (Fig. 2c)[39]. To further understand the occurrence of mitochondrial fragmentation, we assessed the levels of mitochondrial fission and fusion proteins. We observed significant DRP1 upregulation in patients' cells, with no change in OPA1 levels (Fig. 2d), consistent with increased mitochondrial fission process[40]. These findings

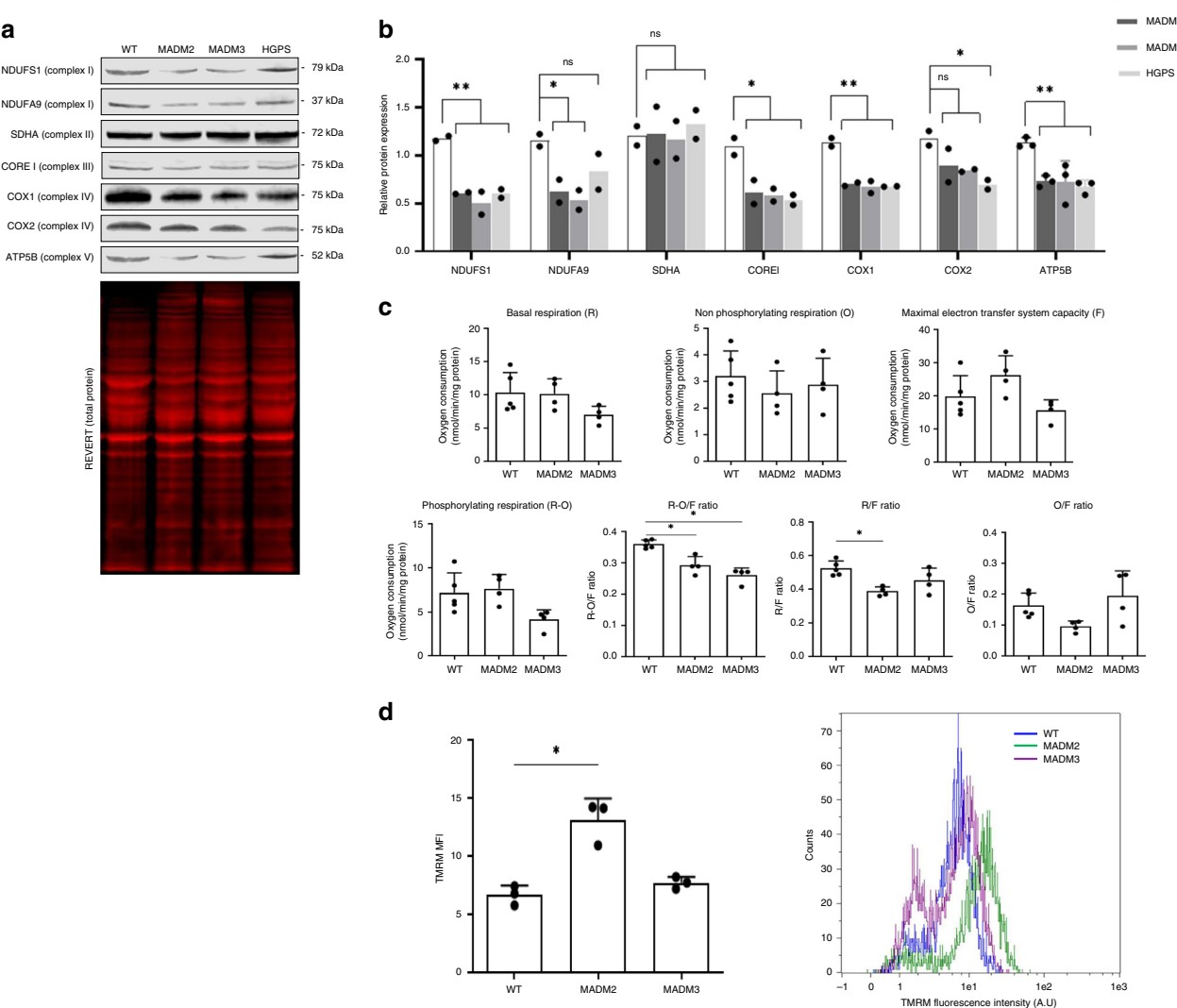

**Fig. 3 Respiratory chain composition and function is impaired in MADaM fibroblasts. a** Representative western blots of respiratory chain proteins of complex I (NDUFS1, NDUFA9), complex II (SDHA), complex III (CORE I), complex IV (COX1, COX2), and complex V (ATP5B) in whole lysates from healthy control (WT), MADM2, MADM3, and HGPS fibroblasts. Protein levels were quantified by ImageJ software and their expression levels were normalized to REVERT total protein. **b** Quantitative analysis of western blots of respiratory chain proteins from complex I (NDUFS1, NDUFA9), complex II (SDHA), complex III (CORE I), complex IV (COX1, COX2), and complex V (ATP5B) from a healthy control (WT), MADM2, MADM3, and HGPS fibroblasts. Protein expression levels were normalized to Revert total protein using ImageJ software. Results are expressed as the mean, for ATP5B the mean ± SD is shown; $n = 3$ independent experiments for ATP5B and $n = 2$ independent experiments for all other proteins. One-way anova was performed comparing patients' values to control (exact $p$ values: NDUFS1: $p_{MADM2} = 0.0034$, $p_{MADM3} = 0.002$, $p_{HGPS} = 0.008$; NDUFA9: $p_{MADM2} = 0.041$, $p_{MADM3} = 0.025$, $p_{HGPS} = 0.140$; SDHA: $p_{MADM2} = 0.954$, $p_{MADM3} = 0.89$, $p_{HGPS} = 0.689$; COREI: $p_{MADM2} = 0.0188$, $p_{MADM3} = 0.0154$, $p_{HGPS} = 0.011$; COX1: $p_{MADM2} = 0.0014$, $p_{MADM3} = 0.0011$, $p_{HGPS} = 0.0012$; COX2: $p_{MADM2} = 0.1129$, $p_{MADM3} = 0.074$, $p_{HGPS} = 0.0257$; ATP5B: $p_{MADM2} = 0.0035$, $p_{MADM3} = 0.0034$, $p_{HGPS} = 0.0016$) ($p$ values: $*p < 0.05$, $**p < 0.01$, ns not significant. **c** Basal (R), oligomycin (O), and FCCP (F) respirations were studied in healthy (WT) and MADM2 and MADM3 fibroblast cell lines at passage P8. Respiratory ratios, including RCR (O/F) and RCRp (R-O/F) are also expressed for the respiration measured at 48 h. Data are expressed as mean ± SD of $n = 4$ independent experiments; statistical significance was analyzed using two-tailed Mann–Whitney test, 95% CI; $*p = 0.015$ in all cases. **d** The graph (left panel) shows quantification of tetramethylrhodamine methyl ester (TMRM, 50 nM) mean fluorescence intensity (MFI) signal measured by FACS analysis (right panel) in control (WT) and MADM2 and MADM3 fibroblasts. The counts shown in the right panel indicate the number of mitochondria quantified in this study, obtained from $\underline{n} = 3$ independent experiments. Data are expressed as mean ± SD; statistical significance was analyzed using two-tailed unpaired Student's $t$ test, 95% CI; $*p_{MADM2} = 0.015$. FACS sequential gating/sorting strategies are provided in Supplementary Fig. 12 and source data are provided as a Source Data file.

were corroborated by the identification of reduced levels of MFN2 (Mitofusin 2, involved in mitochondrial fusion) in the fibroblasts of patients MADM4-2 and MADM4-3 (Supplementary Fig. 5).

Given that reduction of respiratory complex proteins was associated with MTX2 depletion[31] and was also described in HGPS[21], we explored their expression levels in MTX2-deficient fibroblasts (Fig. 3a). Western blot revealed a significantly lower expression of NDUFS1 and NDUFS9 (complex I), CORE1 (complex III), COX1 and COX2 (complex IV), and β-ATPase (complex V), but not of SDHA (complex II) (Fig. 3b). These data prompted the assessment of mitochondrial respiration, which revealed a significant decrease in the respiration dedicated to ATP synthesis (R-O/F) for both patients' fibroblasts (Fig. 3c), together with the basal routine respiration (R/F), and slightly

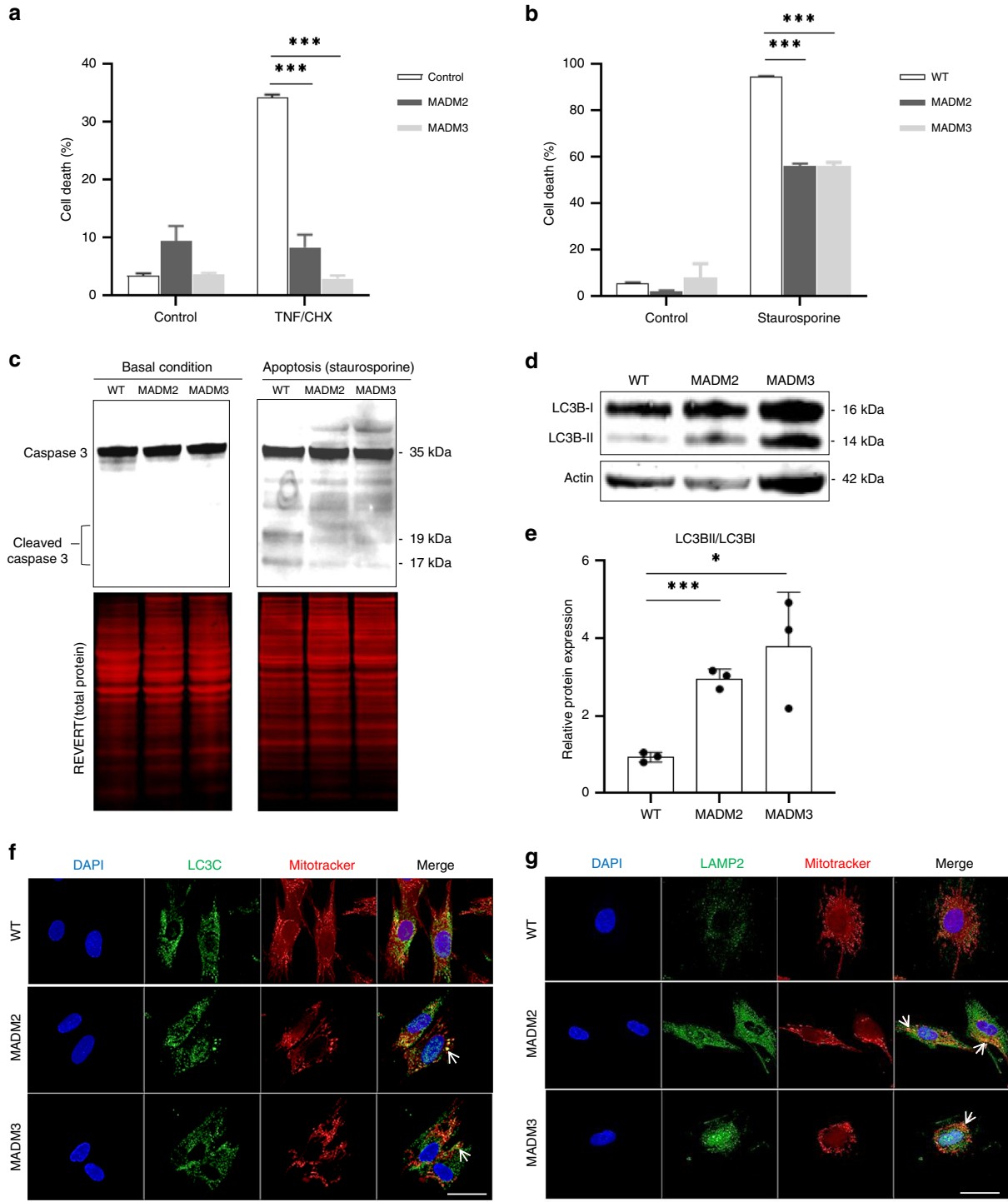

increased mitochondrial membrane potential for patient MADM2 (Fig. 3d). On the other hand, there was no significant difference in reactive oxygen species levels between patients and control fibroblasts (data not shown). These results disclose mitochondrial respiratory chain dysfunction, possibly leading to lower ATP production.

**MTX2 and MTX1 deficiency hampers TNF-α-induced apoptosis.** Given the involvement of MTX2 and MTX1 in apoptosis response to TNFα[33–36], we next investigated whether MTX2 and MTX1 loss could lead to apoptosis resistance in patients'

primary fibroblasts. The number of dead cells upon apoptosis triggering with either staurosporine or TNFα/cycloheximide (CHX) was significantly reduced in patient compared to control fibroblasts (Fig. 4a, b) and confirmed by the reduction of Caspase 3 cleavage in *MTX2*-mutant fibroblasts (Fig. 4c). Given these observations and the inferred obstacle for the patients' organism to sort out altered cells by apoptosis, we next examined whether autophagy and senescence would be increased in patients' fibroblasts, together with reduced proliferation rates, as alternative means of attenuating the noxious effects of the dysfunctional cells.

**Fig. 4 MADaM fibroblasts resist to apoptosis induction and show increased LC3C-dependent mitophagy. a** Control, MADM2 and MADM3 fibroblasts were treated with 100 ng of TNFα and 10 ng of cycloheximide (CHX) for 18 h. Cell death was determined using trypan blue assay. The percentage of dead cells upon drug exposure is shown and error bars represent the SD; $n = 3$ independent experiments. Two-tailed unpaired $t$ test was used to evaluate the statistical significance of differences among the groups (exact p values: $p_{MADM2} = 4.03712E-05$, $p_{MADM3} = 2.12719E-07$); ***$p < 0.001$. **b** Control, MADM2 and MADM3 fibroblasts were treated with 1 μM of staurosporine for 24 h. Cell death was determined using trypan blue essay. The percentage of dead cells upon drug exposure is shown and error bars represent the SD; $n = 3$ independent experiments. Two-tailed unpaired $t$ test was used to evaluate the statistical significance of differences among the groups (exact p values: $p_{MADM2} = 3.70103E-07$, $p_{MADM3} = 0.0004$); ***$p < 0.001$. **c** Fibroblasts were treated with 1 μM of Staurosporine for 6 h and caspase cleavage was determined by western blot. REVERT total protein was used as loading control. One experiment was performed per condition. **d** Immunoblot of LC3B-I and LC3B-II expression in fibroblasts of patients (MADM2 and MADM3) and a control (WT). Protein levels were quantified by ImageJ software and their expression levels were normalized to actin. **e** Quantitative analysis of LC3B-II/ LC3B-I ratio from immunoblots. Protein levels were quantified by ImageJ software and their expression levels were normalized to actin values. Graph bars show the mean ± SD from $n = 3$ independent experiments. Two-tailed unpaired $t$ test was used (exact p values: $p_{MADM2} = 0.0002$, $p_{MADM3} = 0.026$); *$p < 0.05$, ***$p < 0.001$. **f, g** Indirect immunofluorescence staining of LC3C, LAMP2 (green), Mitotracker (red), and DAPI (blue) in patients' (MADM2 and MADM3) and control fibroblasts. Cytoplasmic colocalization of LC3C and LAMP2 with mitochondria can be seen in patients' cells (arrows). $n = 2$ independent experiments were performed for LAMP2 and LC3C staining. Scale bar, 10 μm. Source data are provided as a Source Data file.

The autophagy marker LC3B was studied at basal conditions (Fig. 4d). The LC3B-II/LC3B-I ratio was significantly increased in patients' fibroblasts, indicating increased macro-autophagy in MTX2-deficient cells (Fig. 4e). This process was specifically mediated by the housekeeping mitophagy actor LC3C[41], which colocalized with the fragmented mitochondrial network together with the LAMP2 lysosomal marker (Fig. 4f, g), and not by the classical parkin/ubiquitin autophagic pathway, involving ubiquitin, LC3B and p62 (Supplementary Fig. 7)[42]. A significant increase in doubling times (Fig. 5a) was observed in patients' versus control fibroblasts. A higher percentage of senescence was also confirmed by SA-β-galactosidase quantification, as already reported and observed here for HGPS fibroblasts (Fig. 5b)[18].

**Nuclear morphological changes in MADaM cells resemble HGPS.** Given the clinical and functional similarities with progeroid laminopathies, we assessed nuclear morphologies by DAPI staining, and observed a high percentage of dysmorphic nuclei, again similar to those of HGPS patients (Fig. 5c, d)[17]. Conversely, the assessment of Lamin A/C expression and localization did not evidence differences between control and patient fibroblasts, in which progerin was not expressed (Fig. 5e). In addition, no prelamin A accumulation was detected in patients' cells by immunoblot nor indirect immuofluorescence (Supplementary Fig. 8), as compared to a MADB patient cell line (positive control). Nonetheless, Lamin A/C immunofluorescence analyses revealed profound nuclear morphological abnormalities in MTX2-deficient fibroblasts, including blebs and herniations, previously described in HGPS and other progeroid laminopathies[17,18,43] (Fig. 5f). Interestingly, the MitoTracker and Lamin A/C double staining revealed that the mitochondrial network did not extend to the nuclear rim, as it did in control fibroblasts. These alterations did not involve actin levels or localization, since both were conserved in western blot and indirect immunofluorescence experiments performed in patients' fibroblasts (Supplementary Fig. 9). In addition, since Shumaker et al.[44] and more recently Tian et al.[45], among others, showed that HGPS cells present epigenetic changes, including decreased trimethylation at lysine 9 of histone H3 (H3K9-3me), a constitutive heterochromatin mark, we monitored its levels in MADaM patients' fibroblasts by western blot. No significant variation was found comparing patient to control fibroblasts, while we confirmed a significant reduction of H3K9-3me level in HGPS fibroblasts (Supplementary Fig. 10).

***mtx-2* downregulation in *C. elegans* reproduces nuclear defects.** Because the knockdown of *mtx2*, which is a target of the

pluripotency factor Nanog, leads to early embryonic lethality in zebrafish due to epiboly defects[46], we characterized *mtx-2* depletion in *Caenorhabditis elegans*, a known model to study factors affecting longevity[47], as well as nuclear lamina structure and function[48].

*mtx-2* downregulation was obtained in *C. elegans* using both *mtx-2* RNAi in a strain-expressing lmn-1:gfp[49,50] (Ce-lamin-GFP) (Fig. 5g–h) and *mtx-2* KO in a strain expressing a mitochondrial matrix-targeted GFP in body-wall muscle cells (*mitogfp*) (Supplementary Fig. 11a).

*mtx-2* downregulation by siRNA led to nuclear morphological abnormalities together with Ce-lamin-GFP nuclear aggregates that increased with the worm's age (Fig. 5g–h), similar to previous results on patients' cells (Fig. 5c, d, f).

To check whether complete *mtx-2* inactivation in *C. elegans* recapitulated the mitochondrial fragmentation seen in patients' fibroblasts, we assessed mitochondrial network morphology in body wall muscle cells of *mtx-2* KO worms expressing *mitogfp* (*mtx-2* KO; *mitogfp* strain) and assigned it to either normal ("tubular" and "intermediate") or abnormal ("disordered", "fragmented" and "very fragmented") category (Fig. 5i and methods)[51]. Despite some variability among individual body-wall muscle cells and animals, we observed normal mitochondrial morphology in wild-type *mitogfp* individuals (100%; $n = 17$), consistent with previously published results[52]. In contrast, the majority of *mtx-2* KO individuals expressing *mitogfp* displayed abnormal mitochondrial morphology (74%; $n = 68$, Fig. 5j). Furthermore, while *mtx-2* RNAi did not affect the brood size in the *C. elegans* Ce-lamin-GFP strain (data not shown), we observed that *mtx-2* KO worms exhibited a significant developmental delay: at least 33% of the progeny remained at L2/L3 stage after 44 h at 20 °C, while virtually no animal reached the young adult (YA) stage, compared to 34% in control animals (Supplementary Fig. 11b). The fertility was also reduced in *mtx-2* KO worms, with egg laying rate being only 10% compared to wild-type worms (Supplementary Fig. 11c).

**Wild-type MTX2-cDNA transfection restores disease phenotypes.** Finally, we transfected the human wild-type MTX2-cDNA in patients' MTX2-deficient fibroblasts (Fig. 6a, b) to assess whether it could rescue MTX1 expression and mitochondrial network defects. Indeed, both parameters were consistently restored upon transient transfection (Fig. 6a–d), emphasizing the essential role of MTX2 in maintaining OMM composition and mitochondrial network morphology.

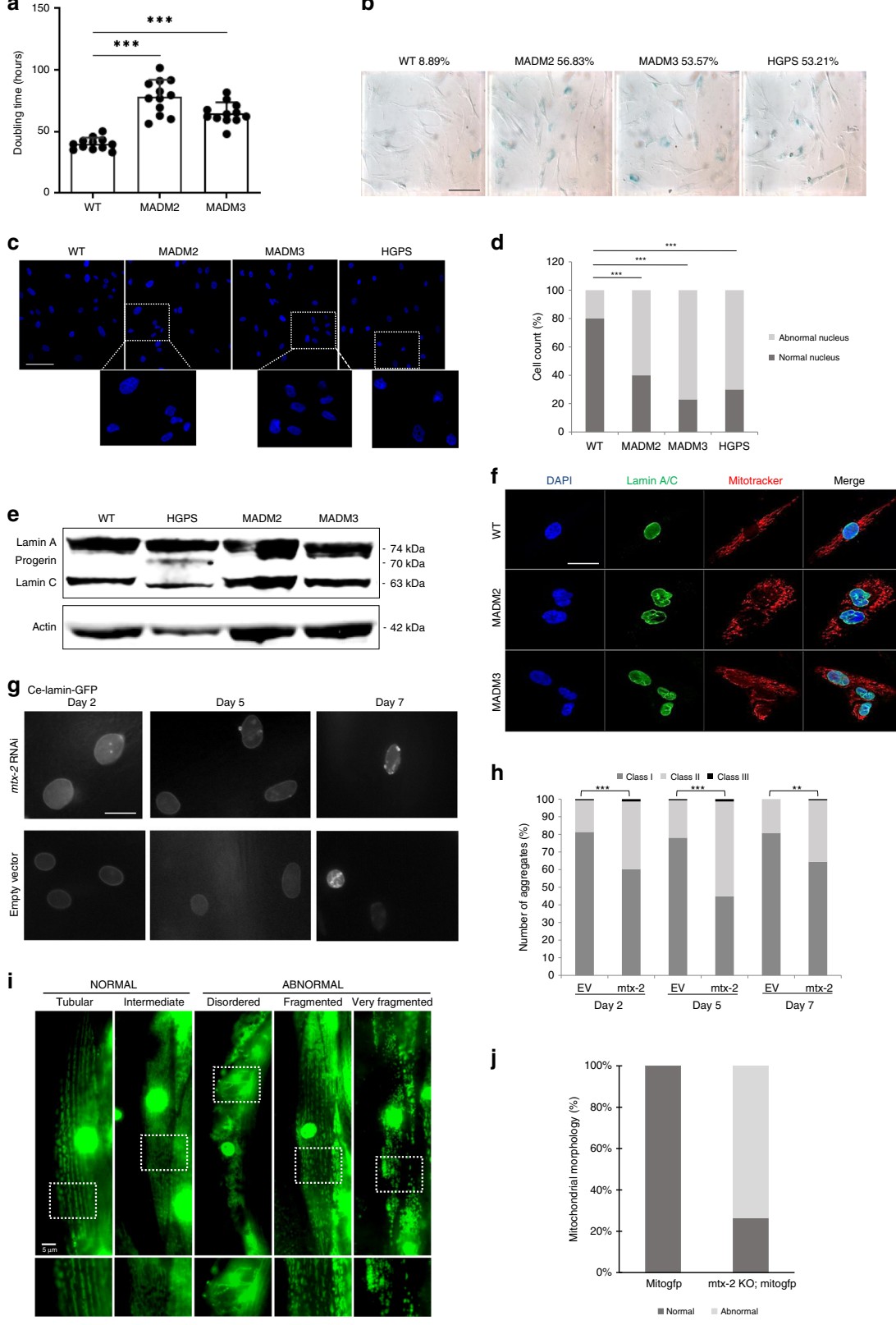

## Discussion

The finding that MTX2 loss causes a MAD syndrome unveils the molecular and cellular bases of a disorder that, originating from the mitochondria, impacts nuclei, leading to a clinical presentation similar to that of Hutchinson–Gilford Progeria, a nucleus-originating disorder. Patients carrying recessive *MTX2* null mutations are functionally equivalent to a double gene KO, leading to absence of the MTX2 protein and also of its partner MTX1. Indeed, since *MTX1* cDNA levels were unchanged, we infer that post-translational MTX1 stability at the OMM was lost due to absence of MTX2. Further experiments aiming to restore MTX1 levels in patients' cells by inhibiting its degradation should

**Fig. 5 MTX2 deficiency induces similar nuclear defects in patients' fibroblasts and *C. elegans* cells. a** Proliferation rates were analyzed by the determination of doubling times in hours ($n = 12$ independent experiments). Results are expressed as mean ± SD. Statistical significance was analyzed using two-tailed Mann–Whitney test (95% CI); ***$p < 0.001$ (exact $p$ values: $p_{MADM2} = 7.39E-07$; $p_{MADM3} = 1.47E-06$). **b** Senescence levels were assessed by SA-β-galactosidase quantification on $n = 225$ (WT), 183 (MADM2), 168 (MADM3) and 171 (HGPS) cells at the same passage number from one experiment. Scale bar, 25 μm. **c** Fluorescence images of nuclei labeled by DAPI staining, showing nuclear morphological abnormalities. Scale bar, 50 μm. **d** Quantification of cells with abnormal nuclear morphology. The average percentage mean values of normal and abnormal nuclei was calculated as described; $n = 150$ independent cells for each cell line at the same passage number were counted from $n = 3$ independent experiment. Chi-square test was used (exact $p$ values: $p_{MADM2} = 3.16E-13$, $p_{MADM3} = 1E-15$, $p_{HGPS} = 1E-15$); ***$p < 0.001$. **e** A representative western blot of fibroblasts' whole lysates showing Lamin A/C, progerin vs. actin expression in control (WT), MADM2, MADM3, and HGPS from one experiment. **f** Immunofluorescence staining of Lamin A/C (green), Mitotracker (red), and DAPI (blue) in patients' and control fibroblasts showing nuclear blebbing and aberrant folding with altered Lamin A/C staining. The images are representative of $n = 3$ independent experiments. Scale bar, 10 μm. **g** *mtx-2* downregulation by siRNA causes nuclear aggregates in *C. elegans*. Wild-type *C. elegans* expressing lmn-1:gfp (Ce-lamin-GFP) were transfected with either *mtx-2* RNAi or empty vectors (EV). Representative images of Ce-lamin-GFP nuclear aggregates monitored at days 2, 5, and 7 after transfection. Scale bar, 10 μm. **h** Graphical representation of the number of nuclear aggregates in aging animals transfected with either empty vector (EV) or *mtx-2* siRNA (mtx-2); $Y$ axis: percent of nuclei in each of three categories (class I: 0 aggregates, class II: 1–5 aggregates, class III: >5 aggregates), $X$ axis: day of adulthood (2, 5, 7) upon transfection; ($n = 145-148$ nuclei were counted from ten independent animals for each condition; average: 14,6 nuclei per worm, cf. Source Data File). Fisher's exact test $p$ values: ***$p_{day2} = 1.14E-04$, ***$p_{day5} = 1.00E-08$, **$p_{day7} = 0.00242572$; *$p < 0.05$, **$p < 0.01$, ***$p < 0.001$). **i** Representative images of the different mitochondrial morphologies observed, subdivided in normal and abnormal as described. Lower panels: magnification of the region surrounded by a rectangle in the upper panels. Scale bar: 5 μm. **j** quantification of mitochondrial morphologies in *mitogfp* strain ($n = 17$ biologically independent animals) and *mtx-2 KO; mitogfp* strain ($n = 68$ biologically independent animals). The data are representative of two independent experiments. Source data are provided as a Source Data file.

validate this point, with potentially important therapeutic consequences. Indeed, based on previous results, MTX1 loss, even more than MTX2 loss, could be a major determinant of apoptosis resistance in patients' cells[36], with downstream "domino" noxious effects. Furthermore, since VDAC2, the third major component of the SAM complex, was not lost upon MTX2 deficiency[31], its presence may be able to rescue MTX1 localization at the OMM upon inhibition of its degradation.

Respiratory chain composition alterations in MADaM patients are very similar to those found in HGPS patients[21], leading to partial OXPHOS dysfunction and inferred lower ATP production. Low levels of intracellular ATP have been previously linked to increased bone-resorbing activity by osteoclasts through Bcl-$x_L$ downregulation[53,54]. It is thus tempting to speculate that this mechanism may play a role in MAD and HGPS specific skeletal pathophysiology, characterized by viscerocranium and distal bone resorption. Deeper studies, exploring calcium handling and ATP production capacity of patients' mitochondria, are thus warranted.

On the other hand, no increased ROS production was observed in patients' fibroblasts, in line with previous results showing that in WT mouse fibroblasts (L929) metaxin deficiency does not increase ROS production[34].

Interestingly, besides being significantly fragmented, mitochondrial network did not reach the nuclear rim in MTX2-deficient cells, suggesting possibly altered cyto-nucleo–skeletal interactions, which may contribute to the observed nuclear deformations[55–58]. Since the cytoskeletal actin network was preserved in patients, the altered mitochondrial network localization may be due to deficient microtubule transport and inefficient juxtanuclear clustering of damaged mitochondria[59]. Further exploration of the different compartments linking the cytoskeleton to the nucleoskeleton should disclose possible "mechanical" links among dysfunctional mitochondria and nuclei in patients' cells.

In this respect, the observation that the downregulation of *C. elegans* mtx-2 by siRNA affects the mitochondrial structure and induces secondary nuclear defects provides a relevant model to study this peculiar pathophysiological mechanism. These defects could lead to cell loss in tissues subjected to mechanical strain or shear stress, as arterial VSMC or the endothelium[60–62], due to altered mechano-transduction signaling pathways[63], possibly

explaining the severe cardiovascular disease observed both in progeroid laminopathies and MADaM.

The striking clinical resemblance among HGPS and MADaM patients is indeed very likely related to commonly altered signal transduction pathways; one hypothesis is that these pathways become activated upon either nuclear or mitochondrial primary defects, reciprocally and secondarily impacting each other. In addition, in both cases, the ubiquitously primary altered organelle, either the nucleus in HGPS, with progerin expression leading to downstream mitochondrial dysfunction[20,21], or mitochondria in MADaM, leading to reduced energy production, impaired apoptosis and nuclear deformation, favor cell senescence and reduced proliferation, converging to a systemic and severe premature aging phenotype[64] characterized by striking clinical similarities and the early demise of affected patients.

Additional explorations of the mitochondrial dysfunctions induced by MTX2 loss should focus on the maintenance of the mitochondrial genome integrity, with respect to the alteration of the mitochondrial network distribution and aggregation, as well as on nuclear genome repair capacities.

Indeed, hampered apoptosis disclosed in MADaM cells may be a primary "metabolic" stress, causing reduced proliferation rates and increased senescence, which in turn may lead to the accumulation of unrepaired DNA lesions and the hyperactivation of a DNA repair checkpoint response, as observed in HGPS[65]. Indeed, nuclear DNA repair defects are the "primum movens" of most known premature aging disorders[18], and it has been shown that in vascular smooth muscle cells (VSMCs) they favor increased p53 signaling and senescence, reduced proliferation and senescence-associated secretory phenotype, causing secretion of inflammatory cytokines, osteoblastic trans-differentiation and ultimately arterial calcification and stiffness, which may underlie MADaM arterial disease, as in HGPS[60].

Nonetheless, some mechanisms diverge in these two disorders, namely prelamin A/progerin accumulation, which is absent in MADaM patients' fibroblasts, and epigenetic changes, namely H3K9-3me levels, which are not reduced in MADaM patients' fibroblasts.

In conclusion, we report a causative link between mutations in *MTX2*, encoding a component of the OMM, and MADaM, a rare and severe progeroid MAD, presenting with many HGPS-like features.

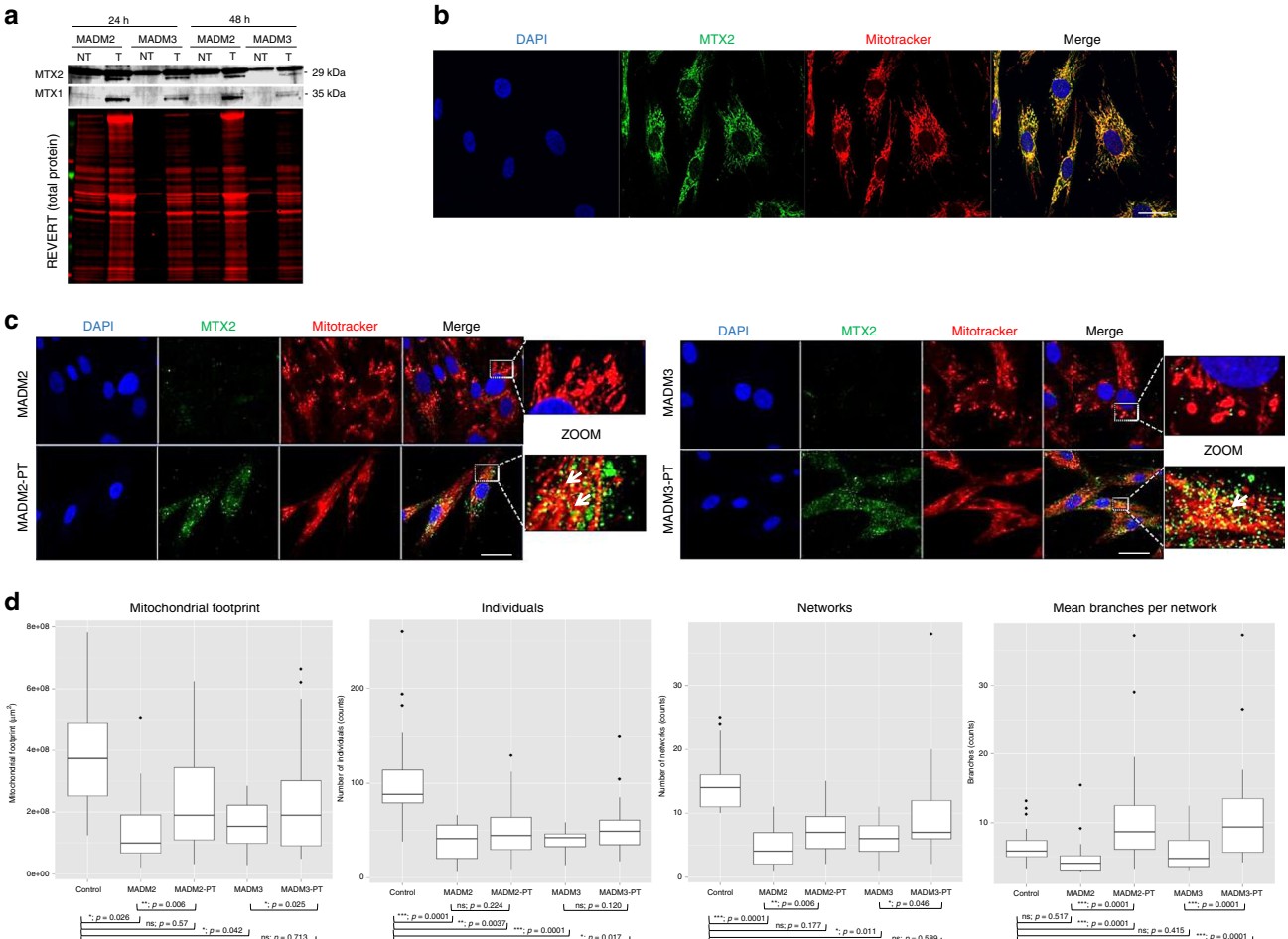

**Fig. 6 MTX2 overexpression by cDNA transfection rescues pathological phenotypes. a** Western blots were performed 24 and 48 h post *MTX2*-cDNA transfection in MADM2 and MADM3 patients' fibroblasts (NT nontransfected, T transfected). The data are representative of one experiment. **b** Immunofluorescence staining of MTX2 (green), Mitotracker (red), and DAPI (blue) in control fibroblasts showing colocalization of MTX2 with mitochondria. The data are representative of one experiment. Scale bar, 10 μm. **c** Immunofluorescence staining of MTX2 (green) and Mitotracker (red) 48 h post *MTX2*-cDNA transfection in MADM2 and MADM3 patients' fibroblasts (PT post transfection). Colocalization of overexpressed MTX2 with mitochondria is shown (arrows in the zoomed images). The data shown are representative of one experiment. Scale bar, 10 μm. **d** Mitochondrial network analysis using the MINA toolset on healthy and MADM2 and MADM3 patients' fibroblasts before and after *MTX2*-cDNA transfection showing improved mitochondrial footprint, individuals, networks, and branches per network for all parameters in PT conditions, when compared either to the patient's nontransfected cells or to control cells. Box plots show median (horizontal lines), first to third quartile (box), and the most extreme values within 1.5 times the interquartile range (vertical lines). Outliers are shown as well. $n = 29$ (WT), 27 (MADM2), 27 (MADM2-PT), 28 (MADM3), and 25 (MADM3-PT) cells from one experiment were blindly scored. Differences between control and patients were analyzed by Kruskal–Wallis test with multiple-comparison and exact *p* values are indicated in the figure. *$p < 0.05$, **$p < 0.01$, ***$p < 0.001$, ns not significant. Source data are provided as a Source Data file.

Importantly, the peculiar apoptosis resistance observed in MADaM patients' cells suggests that novel in vitro models (patients' primary fibroblasts, induced pluripotent stem cells-derived cell lines…) may be used to study frequent age-related morbidities in which defective apoptosis plays a pivotal pathophysiological role, including cancer, immunologic, infectious, neurodegenerative, and cardiovascular diseases[66–69].

## Methods

**Patients.** The explorations on the DNAs of the patients and couples of parents (of MADM1, MADM3 and MADM4-3) were performed first in a diagnostic setting, then in a research setting, complying with the ethical guidelines of the institutions involved and the legislation requirements of the countries of origin. After written informed consents were obtained from the family members, peripheral blood samples from the participants were drawn, and genomic DNA was extracted following standard procedures. Patients MADM2 and MADM3, HGPS, MADB as well as wild-type fibroblasts cell lines explored in this study were prepared, stored, and delivered for research according to the French regulation by the Biological Resource Center (CRB AP-HM Biobank; NF S96-900 & ISO 9001 v2015 Certifications), la Timone Hospital, Marseille, whose cell lines were declared to the

French ministry of Health (Declaration no. DC-2008-429) and whose use for research purposes was authorized by the French Ministry of Education, Research, and Innovation (Authorization nos. AC-2011-1312; AC-2017-2986). Patients MADM4-2, MADM4-3, WT1, and WT2 fibroblasts cell lines were cultured, stored, and delivered for research by Centogene (CENTOGENE AG, Rostock, Germany). All studies were reviewed and approved by Singaporean institutional Review board (NUS IRB 10-051). The authors affirm that the parents of each human research participant provided informed consent for publication of the images in Fig. 1, Supplementary Figs. 1 and 3.

**Whole-genome sequencing.** This study was conducted in collaboration with the Centre National de Génotypage institute (CNG, Paris). Genomic DNA was sequenced on a HiSeq 2000 sequencer according to the manufacturer's instructions (Illumina, San Diego, CA, USA). Clean reads were aligned to the reference human genome (UCSC hg19) using a Burrows–Wheeler Aligner. The average sequencing depth ranged from 35.73× to 38.45×. The mapping rate of clear data ranged from 97.03 to 97.27%, and the genome coverage ranged from 99.83 to 99.85%.

**Whole-genome data analysis.** The results were analyzed using the in-house VarAft software version 2.10[37], which is freely available online. Through genome analysis, both recessive and dominant models of inheritance were studied. Coding

regions were extracted and analyzed using custom scripts. We prioritized rare functional variants (missense, nonsense, splice site variants, and indels) that were homozygous (based on the main hypothesis of an "identity-by-descent" homozygous mutation in each of our patients), heterozygous de novo variants, or compound heterozygous variants that would affect the same gene in the three probands and excluded variants with a minor allele frequency > 0.01 in dbSNP137 and 138, in the Exome Variant Server, 1000 Genomes Project or GnomAD (including Exome Aggregation Consortium (ExAC)) databases. Several online tools were used to predict the functional impact and pathogenicity of the identified variants such as MutationTaster, PredictProtein, PolyPhen, Combined Annotation Dependent Depletion (CADD), SIFT, UMD predictor, and Varsome (URLs in the section below).

Mutation segregation was verified for the parents of patients MADM1, MADM3 (they had been sequenced as well at CNG), and MADM4-3. Sequence variants were described following the Human Genome Variations Society Guidelines available at https://varnomen.hgvs.org/.

**Whole-exome sequencing and data analysis in patient MADM5**. The whole-exome sequencing in patient MADM5 was performed on an Illumina GAIIx Sequencer using the Agilent SureSelect Human All Exon 50 Mb kit, as described in ref. [70]. Approximately 97% of target sequences were covered at least tenfold with a mean coverage of about 75×. Data analysis was performed as described in ref. [70].

**Sanger-sequencing confirmation**. The Primer3 online tool was used to design the specific primers used for PCR amplification and direct sequencing of the genomic region surrounding each variant. Primers used to amplify the genomic mutant exons were as follows:

MTX2-Ex1-F (5′-GGGCTTTGCGAGTCTGAAC-3′), MTX2-Ex1-R (5′-GCCA AGTGTCTCCTTCTCAG-3′), MTX2-Ex4-F (5′-CAGTTTAGAAAAGAAATGCAT AC-3′), MTX2-Ex4-R (5′-CATGGAAAGATTCTTCAAAGGGT-3′), MTX2-Ex6-F (5′-CAAGGTTTGCTGACCATTCTG-3′), MTX2-Ex6-R (5′-GGTAGCCAACTCAA CTCTAGAA-3′), MTX2-Ex9-F (5′-ACTGTGATACAAGTTACGTT-3′), and MTX2-Ex9-R (5′-TGCAACCTAACAGTACAGAA-3′). PCR products were examined by agarose gel electrophoresis and subjected to Sanger sequencing. Whole-genome results and the mutation segregation patterns in the probands and the available parents were confirmed by Sanger sequencing. All primers used to sequence MTX2 coding region (NM_006554.5) are available upon request. Other accession numbers, to which the MTX2 gene (GRCh37/hg19 assembly, chr2:177,134,123-177,202,753) and products can be referred to, are as follows: NCBI CCDS2272.1; Ensembl gene ID: ENSG00000128654; Ensembl Canonical transcript ID: ENST00000249442.10; HGNC:7506.

**Cell culture**. Skin fibroblast cells were grown in Dulbecco's modified Eagle's medium (Life Technologies) supplemented with 15% fetal bovine serum (Life Technologies), 2 mM L-glutamine (Life Technologies), and 1× penicillin–streptomycin (Life Technologies) at 37 °C in a humidified atmosphere containing 5% $CO_2$. Exclusion of mycoplasma contamination was performed on a monthly basis. The experiments were performed on fibroblasts of patients and healthy subjects matched for age and passage number.

**Antibodies**. Antibodies used in this work include: a rabbit anti-MTX2 polyclonal antibody (#HPA031551, used at 1:250 dilution for the western blot analyses, Sigma-Aldrich, Inc.), a rabbit anti-MTX2 polyclonal antibody (#11610-1-AP, used at 1:500 dilution for the western blot analyses, Proteintech®); a rabbit anti-MTX1 polyclonal antibody (#HPA011543, used at 1:250 dilution for the western blot analyses, Sigma-Aldrich, Inc.); a mouse anti-MTX1 monoclonal antibody (#sc-514846, Santa Cruz Biotechnology, Inc.) used at 1:500 dilution for the western blot analyses; a rabbit anti-VDAC2 polyclonal antibody (#HPA043475, used at 1:250 dilution for the western blot analyses, Sigma-Aldrich, Inc.); a mouse anti-VDAC1 monoclonal antibody (#ab14734, used at 1:500 dilution for the western blot analyses, Abcam, Inc.); a rabbit anti-SAMM50 monoclonal antibody (#ab133709, used at 1:500 dilution for the western blot analyses, Abcam, Inc.); a rabbit anti-TOMM40 monoclonal antibody (#ab185543, used at 1:500 dilution for the western blot analyses, Abcam, Inc.); a rabbit anti-TOMM22 polyclonal antibody (#HPA003037, used at 1:500 dilution for the western blot analyses, Sigma-Aldrich, Inc.); a rabbit anti-BAK monoclonal antibody (#ab32371, used at 1:500 dilution for the western blot analyses and at 1:50 for immunofluorescence labeling, Abcam, Inc.); a rabbit anti-BCL2 polyclonal antibody (#ab59348, used at 1:500 dilution for the western blot analyses and at 1:200 for immunofluorescence labeling, Abcam, Inc.); a mouse anti-DRP1 monoclonal antibody (ab56788, used at 1:500 dilution for the western blot analyses, Abcam, Inc.); a rabbit anti-OPA1 polyclonal antibody (#ab42364, used at 1:500 dilution for the western blot analyses, Abcam, Inc.); a rabbit anti-LC3C polyclonal antibody (#ab168813, used at 1:500 dilution for the western blot analyses and at 1:50 for immunofluorescence labeling, Abcam, Inc.); a mouse anti-LC3B monoclonal antibody (#sc-271625 used at 1:1,000 dilution for the western blot analyses and at 1:100 for immunofluorescence labeling, Santa Cruz Biotechnology, Inc.); a rabbit anti-lamin A/C polyclonal antibody which reacts with lamin A, lamin C, and progerin (#sc-20681, used at 1:1,000 dilution for the western blot analyses and at 1:200 for immunofluorescence labeling, Santa Cruz

Biotechnology, Inc.); a mouse anti-progerin monoclonal antibody (#sc-81611 used at 1:1,000 dilution for the western blot analyses and at 1:200 for immuno-fluorescence labeling, Santa Cruz Biotechnology, Inc.); a mouse anti-NDUFS1 monoclonal antibody (#sc-271387, used at 1:500 dilution for the western blot analyses, Santa Cruz Biotechnology, Inc.); a mouse anti-SDHA monoclonal anti-body (sc-390381, used at 1:500 dilution for the western blot analyses, Santa Cruz Biotechnology, Inc.); a mouse anti-COREI monoclonal antibody (#459140, used at 1:500 dilution for the western blot analyses, Invitrogen); a mouse anti-COXI monoclonal antibody (#459600, used at 1:500 dilution for the western blot analyses, Invitrogen); a mouse anti-COXII monoclonal antibody (#ab110258, used at 1:500 dilution for the western blot analyses, abcam, Inc.); a mouse anti-ATP5B mono-clonal antibody (#ab14730, used at 1:500 dilution for the western blot analyses, abcam, Inc.); a rabbit anti-Caspase-3 polyclonal antibody (#9661, used at 1:1000 dilution for the western blot analyses, Cell Signaling, Inc.); a mouse anti-mono- and polyubiquitinated conjugates monoclonal antibody (#BML-PW8810-0100, used at 1:200 for immunofluorescence labeling, Enzo); a mouse anti-actin mono-clonal antibody (#MAB1501R used at 1:5,000 dilution for the western blot analysis, Merck, Inc.); a goat anti-prelamin A polyclonal antibody (#sc-6214, used at 1:1,000 dilution for the western blot analysis and at 1:100 for immunofluorescence labeling, Santa Cruz Biotechnology, Inc.); a rabbit anti-histone H3 (tri-methyl K9) poly-clonal antibody (#ab8898, used at 1:1,000 dilution for the western blot analysis and at 1:500 for immunofluorescence labeling, Abcam, Inc.); a mouse anti-GAPDH monoclonal antibody (#sc-47724, Santa Cruz Biotechnology, Inc.) used at 1/1000 dilution for the western blot analyses; a rabbit anti-SQSTM1/p62 polyclonal antibody (#5114, used at 1:100 for immunofluorescence labeling, Cell Signaling); a mouse anti-Mitofusin 2 monoclonal antibody (#ab56889, abcam, Inc.) used at 1:500 dilution for the western blot analysis.

**RNA isolation and RT-PCR**. Total RNA was obtained using RNeasy plus extraction kit (Qiagen, Valencia, CA, USA) according to the manufacturer's pro-tocol. The extracted RNA was then used for reverse transcription using SuperScript IV Reverse Transcriptase Kit (Applied Biosystems, Waltham, USA). The cDNA obtained was used for the subsequent polymerase chain reaction (PCR). Primer sequences are as follows: MTX2-Ex3-F (5′-GACAATGCAGCTTCTCTTGC-3′), MTX2-Ex7-F (5′-TAGTGATGGGCTGGAGGAAG-3′), MTX2-Ex7-R (5′-AGCTT CATCACACCACTGAA-3′), and MTX2-Ex10-R (5′-TGGTTTTAAAATTCTGAC ACCAA-3′). The reversed transcribed MTX1 mRNA (NM_002455.4) was ampli-fied with these primers (40 cycles) at an hybridization temperature of 59 °C. the PCR-amplified samples were then sequenced to characterize the transcriptional consequences of the homozygous mutations observed in patients MADM2 and MADM3.

**Quantitative RT-PCR studies**. Real-time PCR amplification was carried out with the TaqMan Gene Expression Master Mix (Applied Biosystems) on a LightCycler 480 (Roche, Germany) using predesigned primers for RPS13 (hs-01011487_g1) and MTX1 (Hs00159345_m1) (Thermo Fisher Scientific), using the program: UNG incubation at 50 °C for 2 min, initial denaturation at 95 °C for 10 min, 40 cycles of amplification: denaturation at 95 °C for 15 s and annealing at 60 °C for 1 min. All PCRs were performed in triplicate. Threshold cycle (Ct) values were used to cal-culate relative mRNA expression by the $2^{-\Delta\Delta CT}$ relative quantification method with normalization to RPS13 expression. Transcripts levels' normalization and expression relative to WT was performed using the REST© 2009 software V2.0.13 (Qiagen).

**Western blot**. Total fibroblast proteins were extracted in 200 μl of NP40 Cell Lysis Buffer (Invitrogen, Carlsbad, CA, USA) containing Protease and Phosphatase Inhibitor Cocktail (Thermo Scientific). Cells were sonicated twice (30 s each), incubated at 4 °C for 30 min and then centrifuged at $10,000 \times g$ for 10 min. Protein concentration was evaluated with the bicinchoninic acid technique (Pierce BCA Protein Assay Kit), absorbance at 562 nm is measured using nanodrop 1000 (Thermo Fisher Scientific). Equal amounts of proteins (40 μg) were loaded onto 10% Tris-glycine gel (CriterionTM XT precast gel) using XT Tricine Running Buffer (Bio-Rad, USA). After electrophoresis, gels were electro-transferred onto nitrocellulose membranes or Immobilon-FL polyvinylidene fluoride membranes (Millipore), blocked in odyssey Blocking Buffer diluted 1:1 in PBS for 1 h at room temperature, and incubated overnight at 4 °C or 2 h at room temperature with various primary antibodies. Blots were washed with TBS-T buffer [20 mM tris (pH 7.4), 150 mM NaCl, and 0.05% Tween 20] and incubated with 1:10,000 IR-Dye 800-conjugated secondary donkey anti-goat or IR-Dye 700-conjugated secondary anti-mouse antibodies (LI-COR Biosciences) in odyssey blocking buffer (LI-COR Biosciences). For IR-Dye 800 and IR-Dye 700 detection, an odyssey Infrared Imaging System (LI-COR Biosciences) was used. Total Revert, actin, or GAPDH were variably used as internal protein loading controls.

**Indirect immunofluorescence**. For immunostaining, cells were fixed with 4% paraformaldehyde (PFA), washed with PBS, and permeabilized with 0.5% Triton X-100 for 15 min. After PBS washing, coverslips were incubated with 1% bovine serum albumin for 30 min at room temperature before adding the primary anti-bodies for 3 h at 37 °C or overnight at 4 °C. After washing, the cells were then

incubated with secondary antibodies (A11001, A11058, Life Technologies; 1/400) for 1 h at room temperature. For actin staining, Phalloidin CruzFluor™ 488 Conjugate (sc-363791, Santa Cruz) was diluted at 1× and added to the wells for 1.5 h at room temperature. Then the wells were rinsed for 5 min with DPBS, three times. Nuclei were stained with DAPI (50 ng/ml) diluted in Vectashield (Abcys) for 10 min at RT. The stained cells were observed with a Zeiss LSM 800 Confocal Microscope using Zen 2.3 software (Zeiss, Germany).

**Quantification of mitochondrial networks abnormalities**. Cells were grown on glass coverslips and stained by incubation with 200 nM MitoTracker stain (Molecular Probes) in cell culture medium for 30 min at 37 °C. Samples were washed with phosphate-buffered saline (PBS), fixed in 4% PFA, and analyzed with a Zeiss LSM 800 Confocal Microscope using Zen 2.3 software (Zeiss, Germany). Images obtained from cultured patients and controls cells were processed using the Mitochondrial Network Analysis (MiNA) toolset, a combination of different ImageJ macros that allows the semi-automated analysis of mitochondrial networks in cultured mammalian cells[39]. Briefly, the image was converted to binary by thresholding following the conversion to a skeleton that represents the features of the original image using a wireframe of lines of one pixel wide. All pixels within a skeleton were then grouped into three categories: end point pixels, slab pixels, and junction pixels. The plugin analyzes how the pixels are spatially related and is defined to measure the length of each branch and the number of branches in each skeletonized feature as well as the mitochondrial network morphology. The parameters used in the study were (1) individuals: punctate, rods, and large/round mitochondrial structures; (2) networks: mitochondrial structures with at least a single node and three branches; (3) the mean number of branches per network; and (4) the mitochondrial footprint, which represent the area occupied by mitochondrial structures and which is calculated from the binarized image prior to skeletonizing.

**Measurement of senescence**. Senescence was examined in cells by a senescence-associated β-galactosidase assay following the manufacturer's protocol (Cell Signaling #9860). Cells were seeded in four chamber-well slides (SPL Lifesciences, Korea), washed with PBS and fixed in Fixative solution (1/10 dilution) for 15 min at RT. Cells were washed in PBS, and stained overnight at 37 °C with SA-β-galactosidase staining solution. Stained samples were visualized using a bright field microscope (Leica, Wetzlar, Germany).

**Cell death measurement**. Controls and patients' cells were seeded in six-well plates (50,000 cells/well) and treated either with 1 μM Staurosporine (Sigma-Aldrich) for 24 h or with 100 ng/ml of recombinant human TNF-alpha + CHX (10 ng/ml) for 18 h. The number of alive and dead cells was determined by the trypan blue exclusion test. The Countess™ Automated Cell Counter (Invitrogen) was used to count the number of alive and dead cells after trypan blue staining.

**Proliferation assay**. Fibroblasts were seeded at 20,000 cells per well, and their shape and density were observed by photonic microscopy in standard growing condition. Cell proliferation was assessed by the The Incucyte ZOOM System (Essen BioScience). Percentage of confluency was followed for 96 h using the Basic Analyzer phase confluence software and the doubling time inferred during exponential phase using the $(t2 - t1) \times \ln2/((\ln t2) - \ln (t1))$ formula.

**Abnormal nuclear morphology quantification**. Fibroblasts from patients and controls were cultured with DMEM medium. Cells stained with DAPI were examined by fluorescence microscopy with an Axioplan 2 imaging microscope (Zeiss, Germany). The quantification of cells with abnormal nuclear morphology was performed using the Nuclear Irregularity Index plugin of the ImageJ software (version 1.6.0, NIH, USA). Nuclei were divided in two classes: normal nuclei (nuclei with a smooth oval shape) and abnormal nuclei (nuclei with blebs, irregular shape, or multiple folds) and the respective percentages were calculated in three independent experiments. The results are expressed graphically as the average percentage of the total nuclei counted. At least 200 fibroblast nuclei were randomly selected for each cell line in each experiment.

**Mitochondrial respiration**. Mitochondrial respiration was measured on intact cells resuspended in DMEM-F12 at 37 °C in a high-resolution oxygraph (Oroboros, Innsbuck, Austria). Basal, coupled, maximal, and residual respirations were measured in the absence of exogenous substrate, after the addition of 4 μg/ml oligomycin (O4876, Sigma-Aldrich), after the addition of 1 μM FCCP (C2920, Sigma-Aldrich), and after the inhibition of complex I and III, using rotenone at 5 μM (R8875, Sigma-Aldrich) and antimycin-A at 2 μg/ml (A8674, Sigma-Aldrich), respectively. A total of $4$–$5 \cdot 10^6$ cells were added in the oxygraphic chamber and the analysis started with routine respiration (R) measurement, which is defined as the basal respiration without additional substrate or effector. Then, the F0F1-ATP synthase was inhibited with oligomycin (4 μg/ml), to measure the non-phosphorylating respiration (O), from which was deduced the phosphorylating respiration (R–O). Then the uncoupled oxidative phosphorylation, or maximal endogenous respiration, (F) was measured by stepwise titration of FCCP (carbonyl cyanide p-trifluoromethoxyphenylhydrazone,

0.6–1.2 μM) up to the optimum concentration. The part of the maximal capacity used for oxidative metabolism was calculated as R/F and the part of the maximal capacity used for oxidative ATP synthesis was calculated as (R–O)/F. Finally, respiration was inhibited by rotenone and antimycin-A (doses detailed above). The part of the nonphosphorylating respiration was inferred by calculating the O/F ratio. All experiments were performed at least twice on independent cell cultures.

**ROS production**. Fibroblasts were seeded at 30,000 cells per well and incubated with 5 μM CellROX Green Reagent (C10444, Life Technologies—Thermo Fisher Scientific) for 30 min (37 °C, 5% CO$_2$). Incubation with 250 μM Pyocyanine (P0046, Sigma-Aldrich) for 1 h before analysis provided a positive control. Cells were rinsed with DMEM media, collected and analyzed by flow cytometry MACSQuant (Miltenyi). The Macsquantify software was used to calculate the mean fluorescence intensity (MFI) of the CellROX green reflecting the ROS production.

**Membrane potential measurement**. Fibroblasts were seeded at 30,000 cells-per well and incubated with 0.1 μM tetramethylrhodamine, methyl ester, perchlorate, or TMRM (T668, Life Technologies—Thermo Fisher Scientific) and 5 μM MitoTracker Green (M7514, Invitrogen-Thermo Fisher Scientific) for 30 min. Incubation with 10 μM FCCP (C2920, Sigma-Aldrich) for 30 min was used as a negative control. Cells were rinsed with DMEM media, collected and analyzed by flow cytometry MACSQuant (Miltenyi). The Macsquantify software was used to calculate the MFI of the TMRM green reflecting the membrane potential.

***C. elegans* studies upon *mtx-2* downregulation by siRNA**. A *C. elegans* N2 (wild-type) strain expressing lmn-1:gfp[48] (PD4810 strain expressing Ce-lamin-GFP) was used for *mtx-2* RNAi. Animals were kept at 23 °C, maintained and manipulated under standard conditions as described in ref. [49]. Worms were grown on feeding plates containing 1 mM IPTG, 50 μg/ml ampicillin and seeded with *E. coli* HT115 strains harboring the L4440 empty vector (EV) as a control or specific RNAi vectors, at the appropriate developmental stage. For the classification of nuclear morphology, animals were synchronized by bleaching and were placed on feeding plates seeded with EV or specific RNAi vector. Animals were transferred to a fresh RNAi plate every 48 h to exclude the new progeny. Morphological classification began after 2 days on RNAi plates. Ce-lamin-GFP nuclear aggregates were monitored directly using GFP fluorescence. Multiple images were taken from the middle part of the worm (excluding the head, the tail and the gonads). Nuclei were grouped into three different classes (I–III) based on the number of aggregations foci observed; 0 aggregations: class I, 1–5 aggregations: class II, and more than 5 aggregations foci: class III. The nuclei counted were derived from the hypodermis and neuronal nuclei were excluded from the counting. 145 to 148 nuclei were counted from ten biologically independent animals for each condition (average: 14,6 nuclei per worm). The Fisher's exact test (https://www.danielsoper.com/statcalc/calculator.aspx?id=58) was used for statistical comparison of the number of nuclear aggregates observed in EV vs *mtx-2* siRNA transfected worms each day.

***C. elegans* studies upon *mtx-2* gene knock-out**. *C. elegans* strains were maintained and cultured as described in ref. [50]. The *C. elegans* strain inactivated for *mtx-2* used in this study was *mtx-2(gk444)*, backcrossed three times to wild-type N2 to eliminate other eventual mutations. The *C. elegans* strain CB5600 (ccls4251(Pmyo-3::Ngfp-lacZ; Pmyo-3::Mtgfp) I;him-8(e1489) IV) had GFP fusion proteins localized to mitochondria and nuclei[51] in body wall muscle cells and was named *mitogfp* in this article. The transgenic strain expressing *mitogfp* in a *mtx-2(gk444)* KO homozygous background was generated in this study by conventional crosses and was named *mtx-2* KO; *mitogfp*. PCR amplification was performed to validate genotypes (Supplementary Fig. 11a) using forward (5′-ATCACAGGGTTCAAC GCCAT-3′) and reverse (5′-ACCGAAAATGTTTGATTGTTTCG-3′) primers specific to *C. elegans mtx-2* (NCBI Gene ID: 176087; RefSeq NM_066288.4). The PCR products of either 657 or 132 bp were amplified before and after gk444 deletion, respectively. Transgenic 2-day-old adults were mounted on a 2% agarose pad on a glass slide with 10 μL drop of polystyrene beads solution and analyzed by an inverted epifluorescence microscope (AxioObserver.Z1, Zeiss, Germany) equipped with a LED lamp (pE-300 white, CoolLED) for fluorescence excitation. Fluorescent images were taken using a monochrome-cooled sCMOS camera (ORCA-Flash 4.0 LT, Hamamatsu) with filter set No 38 for green emission at the Plan-Apochromat 63× NA 1.4 objective (Zeiss, Germany). Experiments were automated using Zen software (Zen 2 blue edition, Zeiss, Germany). Qualitative assessment of mitochondrial morphology in body wall muscle cells was performed on the images based on morphological categories as follows: normal: a majority of long interconnected mitochondrial networks (tubular) or a combination of interconnected mitochondrial networks along with some smaller fragmented mitochondria (intermediate); abnormal: images containing a fusion of mitochondrial networks (disordered), a majority of short mitochondria (fragmented) or sparse small round mitochondria (very fragmented). The experiment was repeated twice at different occasions.

**Development and fertility of *mtx-2(gk444)* KO worms**. To determine the developmental features of the *mtx-2(gk444)* KO worms, synchronized L1 stage worms (N~50) were placed onto NGM seeded with OP50 (in triplicate). The larval

development of L1 worms was monitored and identified after 44 h at 20 °C. Developmental stages were identified under stereoscope following size/vulva present in the uterus. L2 were small and L3 were bigger, more mobile and displaying a pre-vulva space. L4 were identified by the appearance of the premature vulva, whereas YAs had a fully formed mature vulva. Results are presented as worms in each developmental stage as a mean percentage of the total population size (Supplementary Fig. 11b).

To determine the egg laying rate, synchronized L4 worms were transferred to a plate overnight at 20 °C. Fifteen YAs with 5–8 eggs present in their uterus were picked onto individual plates for 3 h at 20 °C (in triplicate). The adult worms were removed from the plates and the number of laid eggs were counted. Plates with eggs were incubated for 3 more days to allow hatching larvae counted. Results are presented as the number of laid eggs and hatched larvae per worm per hour (Supplementary Fig. 11c).

**cDNA transfection assay**. For transient transfections, cells were grown up to 50% confluence in 100-mm-diameter dishes in DMEM containing 15% fetal bovine serum and transfected with 6 μg of MTX2-cDNA using jetPRIME™ (Polyplus-transfection Inc., Illkirch, France) according to the manufacturer's protocol. Transfected cells were then incubated for an additional 48 h before downstream processing. The cDNA transfected was human wild-type *MTX2* (Myc-DDK-tagged) encoding *MTX2* transcript 1 (RC209825, CliniSciences).

**Statistics and reproducibility**. All quantifications were performed unblinded. Statistical parameters including the definitions and exact value of $n$ (e.g., total number of experiments, animals, cells, $p$ values, and the types of the statistical tests are reported in the figures and corresponding figure legends. Comparisons between groups were planned before statistical testing and target effect sizes were not predetermined. Statistical significance was analyzed using unpaired Student's $t$ test or two-tailed Mann–Whitney test for two groups unless otherwise stated in the figure legend. $p$ values: $*p < 0.05$, $**p < 0.01$, and $***p < 0.001$ were considered significant. Data shown in column graphs represent mean ± standard error of the mean (SEM) or mean ± standard deviation (SD), as indicated in the figure legends, and individual data points are plotted whenever possible. Statistical analysis was conducted on data from three or more biologically independent experimental replicates excepted when otherwise stated. Statistical analyses were performed using R Statistical Software, Microsoft Excel or Prism 7 (GraphPad Software). The source data for statistical testing can be found in the source data file.

**URLs**. VarAFT software, http://varaft.eu/index.php; Exome Variant Server, http://evs.gs.washington.edu/EVS/; 1000 Genomes Project, http://www.1000genomes.org/; Exome Aggregation Consortium database (ExAC), Cambridge, MA, http://exac.broadinstitute.org; GnomAD, https://gnomad.broadinstitute.org/gene/; Online Mendelian Inheritance in Man (OMIM), http://www.omim.org/; UCSC Genome Browser, http://genome.ucsc.edu/; National Center for Biotechnology Information (NCBI), https://www.ncbi.nlm.nih.gov/; MutationTaster, http://www.mutationtaster.org/; PredictProtein, https://www.predictprotein.org/; PolyPhen, http://genetics.bwh.harvard.edu/pph2/; Combined Annotation Dependent Depletion (CADD), http://cadd.gs.washington.edu/; SIFT, http://sift.bii.a-star.edu.sg/; UMD predictor, http://umd-predictor.eu/; Varsome, https://varsome.com/; Primer3, http://bioinfo.ut.ee/primer3-0.4.0/. REST 2009 software V2.0.13 (Qiagen), https://www.gene-quantification.de/rest-2009.html.

**Reporting summary**. Further information on research design is available in the Nature Research Reporting Summary linked to this article.

## Data availability

The authors state that all data generated during this study are included in the article, and that they are available from the corresponding author upon reasonable request. DNA sequence data that support the findings of this study have been deposited in GenBank (NCBI) under accession numbers: MN255812, MN255813, MN255814, MN255815 and MT124664. *MTX2* variants informations have been deposited at ClinVar: #SUB6063273 and #SUB6975375. Source data are provided with this paper.

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

## Acknowledgements

We warmly acknowledge all family members and their relatives for their participation to this study. Professor Raoul Hennekam is warmly acknowledged for addressing us two of the patients, upon genetic counseling and Professor Patrice Roll for useful discussions. This work was supported by the Association Française contre les Myopathies (AFM) (AFM grant MNH-Decrypt 2011-2015 and TRIM-RD 2016-2020 to A. D.S.G. and N.L.), the Institut National de la Santé et de la Recherche Médicale (INSERM) (recurrent grants) and Aix-Marseille University (AMU) by the RARE-MED Amidex project (A*MIDEX—AAP Méditerranée 2014: RARE-MED Project: "Setting up a Mediterranean Research Network for the study of rare diseases in the Mediterranean area"); a grant by Deutsche Forschungsgemeinschaft (LE4223/1-1) to D.L. Whole-genome sequencing was performed by the Centre National de Génotypage (CNG, Paris: Steven McGinn, Anne Boland and Robert Olaso are warmly acknowledged for their contribution), financially supported by the GENMED Laboratory of Excellence on Medical Genomics, Agence Nationale de la Recherche (ANR-10-LABX-0013). We deeply thank Pr. E. Tournier-Lasserve and her team (INSERM UMR 1141 research unit, Lariboisière Hospital), for allowing CeleScreen's team a free access to molecular biology equipment and facilities. We thank Maylis Raymond (CeleScreen SAS) for excellent technical assistance, Professor Nathaniel Szewczyk (University of Nottingham) for providing CB5600 strain and technical advice on *C. elegans* mitochondrial morphology experiments, the ImagoSeine facility, member of the France BioImaging infrastructure supported by the French National Research Agency (ANR-10-INSB-04, «Investments for the future») for imaging facility and technical support. Some strains were provided by the CGC, which is funded by NIH office of Research Infrastructure Program (P40 OD010440).

## Author contributions

G.B., S.N., H.K., N.A.M., L.S., A.L.P., and R.R. carried out the clinical characterization and follow up of the patients; Bioinformatics analyses were performed by S.E. on patients MADM1 to 3, upon the supervision of A.D.S.G., upon data production from J.F.D.; P.B. and C.P. carried out exome and Sanger segregation on patients MADM4 at CENTOGENE; D.L. and C.K. supervised molecular genetics analyses in patient MADM5; S.E. performed and interpreted most functional analyses on patients' fibroblasts, with the contribution of K.H., C.B., C.A. and N.E.B. respectively under the supervision of A.D.S.G. and B.R; M.L.M. performed and analyzed proliferation assays, mitochondrial respiration, ROS production and membrane potential measurement assays, under the supervision of G.L.; A.R. contributed to protein analysis of components of the mitochondrial respiratory chain; C.C. performed and analyzed the results of functional experiments of siRNA-induced *mtx-2* downregulation in *C. elegans*, supervised by Y.G.; L.M. and S.H.L. performed and analyzed the results of mitochondrial morphology in *mtx-2* KO expressing *mitogfp* in *C. elegans* together with developmental and fertility studies; A.M. and N.L. contributed useful discussions for interpretation of the data. A.L. contributed as a Research Officer. S.E. and A.D.S.G. wrote the manuscript with contributions from G.B., S.N., H.K., N.E.B, A.M., Y.G., P.M., N.L., G.L., and B.R.; all authors read and approved the final manuscript.

## Competing interests

The authors declare no competing interests.

## Additional information

Sahar Elouej[1,21], Karim Harhouri[1,21], Morgane Le Mao[2], Genevieve Baujat[3], Sheela Nampoothiri[4], Hülya Kayserili[5], Nihal Al Menabawy[6], Laila Selim[6], Arianne Llamos Paneque[7], Christian Kubisch[8], Davor Lessel[8], Robert Rubinsztajn[9], Chayki Charar[10], Catherine Bartoli[1], Coraline Airault[1], Jean-François Deleuze[11], Agnes Rötig[12], Peter Bauer[13], Catarina Pereira[13], Abigail Loh[14], Nathalie Escande-Beillard[14], Antoine Muchir[15], Lisa Martino[16], Yosef Gruenbaum[10], Song-Hua Lee[16], Philippe Manivet[16,17,18], Guy Lenaers[2], Bruno Reversade[14], Nicolas Lévy[1,19] & Annachiara De Sandre-Giovannoli[1,19,20]✉

[1]Aix Marseille Univ, INSERM, MMG, U1251, Marseille, France. [2]MitoLab, Mitochondrial Medicine Research Centre, Institut MITOVASC, CNRS UMR 6015, INSERM U1083, Université d'Angers, CHU d'Angers, Angers, France. [3]Department of Medical Genetics, INSERM UMR 1163, Paris Descartes-Sorbonne Paris Cité University, IMAGINE Institute, Necker Enfants Malades Hospital, Paris, France. [4]Department of Pediatric Genetics, Amrita Institute of Medical Sciences & Research Centre, AIMS Ponekkara PO Cochin, Kerala, India. [5]Medical Genetics Department, Koç University, School of Medicine (KUSoM), Istanbul, Turkey. [6]Neurology and Metabolic Division, Cairo University Children Hospital, Cairo, Egypt. [7]Medical Genetics Service Specialties Hospital FF AA No.1, Quito, Ecuador. [8]Institute of Human Genetics, University Medical Center Hamburg-Eppendorf, Hamburg, Germany. [9]Pole of Anesthesiology and Reanimation, Necker Enfants Malades Hospital, Paris, France. [10]Department of Genetics, Institute of Life Sciences, Hebrew University of Jerusalem, Jerusalem, Israel. [11]Centre National de Recherche en Génomique Humaine (CNRGH) and Centre d'Etude du Polymorphisme Humain (CEPH), Institut de Biologie François Jacob, CEA, Université Paris-Saclay and Fondation Jean Dausset, Paris, France. [12]INSERM UMR1163, Institut Imagine, Paris, France. [13]CENTOGENE AG, Rostock, Germany. [14]Institute of Medical Biology, A*STAR, Singapore, Singapore. [15]Sorbonne Université, INSERM, Institute of Myology, Center of Research in Myology, Paris, France. [16]CeleScreen SAS, Paris, France. [17]APHP, Biobank Lariboisière BB-0033-00064, Platform of BioPathology and Innovative Technologies in Health, Hôpital Lariboisière, Paris, France. [18]INSERM UMR1141 « NeuroDiderot », Université de Paris, Paris, France. [19]Department of Medical Genetics, La Timone Children's Hospital, Marseille, France. [20]Biological Resource Center (CRB-TAC), Assistance Publique Hôpitaux de Marseille, La Timone Children's Hospital, Marseille, France. [21]These authors contributed equally: Sahar Elouej, Karim Harhouri. ✉email: annachiara.desandre-giovannoli@univ-amu.fr

