## [Peer Review File · Nature Communications]

Reviewers' comments first round:

Reviewer #1 (Remarks to the Author):

In the article entitled "Loss of MTX2, responsible for a novel severe progeroid mandibuloacral dysplasia, links mitochondrial dysfunction to altered nuclear morphology" the authors report the identification of new null mutations in MTX2 leading to a novel severe premature aging syndrome, named MADaM. Furthermore, they have characterised at a cellular level, both structure and metabolism alterations, due the loss of MTX2.

The findings reported in this article are clear and the choice of the performed experiments is in line with the overall purpose of the study. However, there are several concerns that need to be addressed by the authors.

Major:

The introduction looks more like an abstract. We invite the authors to use that part of the article as abstract and write an appropriate introduction. We suggest moving lines 70-77 and 106-116 from the results section to introduction with a suitable integration of the missing parts. For a better understanding, it would be helpful for the reader to have the results section divided into paragraphs.

The discussion/conclusion paragraph is missing. We suggest to use lines 217-235 as discussion paragraph. Lines 217/219 should be at the end of the discussion as future perspectives. The sentence "Furthermore, MADaM may become a useful model to study frequent age-related disorders..." (lines 235-237), is highly speculative and lacks of appropriate introduction and a deeper discussion. The reader is left wondering about the possible connections between the described mutations with cancer, neurodegenerative and cardiovascular diseases. Please, support your speculations with evidences.

Why MADaM is correlated with age related disorders? We agree that the referred phenotype "...converge to a systemic premature aging phenotypes" but a deeper explanation will facilitate the reader. Please provide direct evidences in both introduction and discussion.

The absence of MTX2 result into a lack of expression of MTX1, even if the MTX1 is not affected (data not shown). It would be nice to have evidences about the possible transcriptional (qPCR on MTX1 cDNA) or translational (proteasome inhibition and confirmation of MTX1 presence or its ubiquitination) regulation of MTX1 transcript.

Since the authors suggest that MTX2 absence impairs cyto-nucleo-skeletal network (line 181-185), we wonder why actin is used as reference protein for western blot analysis. We suggest using GAPDH or another well-characterised reference protein. It would be nice, since it is evident by the WB in fig. 2a that actin is differentially expressed by WT and MADM2 and MADM3 (MTX1 panel), to have immunofluorescence images of the cytoskeleton of patients' fibroblasts. Moreover, lines 181-185, mainly speculative, should be moved in the discussion section (otherwise cyto-nucleo-skeletal interactions alteration should be proved within the article).

Regarding the WB analysis, we found that the representative images used do not reflect the journal standards, therefore we invite the authors to choose better ones. Lastly, we are wondering why for each tested protein the authors show the actin. Is it not part of the same membrane? Do all the proteins have the same MW? Please explain.

All immunofluorescence images have a too low resolution, the same applies also to all the figures (supplementary fig. 3 is impossible to read). Please provide images with higher dpi (300 to 600). The authors provide evidences of severe mitochondrial impairment upon MTX2 absence, it could be interesting and more complete whether the authors could integrate mitochondrial respiration and potential data with calcium measurements.

Minor:

Line 60: Mandibuloacral

Line 88: "...by pairing the patients on a phenotypic basis." Please explain.

Line 97: "...were found heterozygous for the mutation,...."

Line 115-116: ...triggered...

Line 120-123: remove by RT-PCR. If it is sequence analysis is better to refer to sequencing. Please clarify when you talk about cDNA mutation or genomic alterations (same also in sup. Fig. 3)

Line 127-129: The sentence is intricate. Please rephrase.

Line 155: Please remove known.

Line197-198: "...confirming that MTX2-deficient..." This sentence is speculative therefore should be moved to the discussion.

Line 442: consanguineous.

Line 454: WB quantification should be done conservatively: please normalize or on REVERT or on reference protein (no actin, see comment above). Otherwise, give a valid explanation.

Figure 2: It is not clear if the statistic described in line 465-468 apply only to panel d or also to panel a. Which is the n for the WB of panel a?

Line 480: "...Student's t test was performed comparing patients' value to control."

Figure 4: Please number the panels from left to right, top to bottom.

Figure 5n: The figure selected are not clear. Please provide better ones.

Supplementary information Line 46: progressive progressive

Supplementary information Line 268: Which counting methods was used for trypan blue exclusion test?

Supplementary fig. 3a: Alignment with *C. elegans* sequence is missing. *Xenopus* l.

Supplementary fig. 3b: RT-PCR refers to reverse transcription and not to PCR amplification. Please refer to PCR in the legend. What does T stand for?

Supplementary fig. 3c: The introns numbering is not clear: c.208...should be on intron 4 and c.544 on intron 9. Please name the mutations in the same way.

Supplementary fig. 4: Please use better images.

Supplementary fig. 5: Please use a better resolution.

Carlotta Giorgi

Reviewer #2 (Remarks to the Author):

The major claims of the paper submitted by the corresponding author Annachiara De Sandre-Giovannoli are that a form of Mandibuloacral Dysplasia is associated with homozygous null mutations in a mitochondrial membrane gene, *MTX-2*, encoding for the outer mitochondrial membrane protein metaxin-2 and the pathogenetic effect of the mutation is linked altered morphofunctional organization of mitochondrial network, loss of cellular response to apoptotic stimuli, activation of an unconventional autophagic pathway and nuclear morphological defects. This is the first paper linking Mandibuloacral dysplasia to mitochondrion dysfunction, although impaired mitochondrial activity has been observed in the most severe progeroid laminopathy, Hutchinson-Gilford progeria and in a familial partial lipodystrophy associated with LMNA mutations, the FPLD2. The results are mostly unexpected and novel and, since Mandibuloacral dysplasia is a progeroid syndrome with partial lipodystrophy and bone resorption, they will be of interest to researchers in the field of progeroid syndromes and in the wider field of normal and pathological ageing research, research on adipose tissue dysfunction and bone turnover disorders.

A few interesting points need to be addressed:

- The authors nicely show that nuclear morphology is altered, while lamin A/C is not affected. Based on published data, which need to be cited (Filesi et al 2005; Lombardi et al., 2007; Camozzi et al 2012), prelamin A accumulation could be involved. Confocal microscopy analysis of prelamin A immunostaining with diverse antibodies is needed to rule out or acknowledge this hypothesis.
- An important paper shows that mitochondrial dysfunction interferes with chromatin epigenetic modifications and directs longevity-associated mechanisms (Tian et al, Mitochondrial Stress Induces Chromatin Reorganization to Promote Longevity and UPRmt, Cell 2016). This paper must be discussed and chromatin marks as H3K9-dimethyl need to be analyzed, also considering that H3K9 epigenetic defects have been described in Mandibuloacral dysplasia (Filesi et al., 2005; Camozzi et al., 2012).

Reviewer #3 (Remarks to the Author):

Confirming the genetic heterogeneity linked to progeroid syndromes, the authors reported four homozygous null mutations in MTX2 gene in six patients presenting a novel severe premature aging syndrome characterized by growth retardation, bone resorption, arterial calcification, renal glomerulosclerosis and severe hypertension. This new progeroid gene encodes for an outer mitochondrial membrane protein and in vitro functional studies demonstrated that loss of MTX2 leads to mitochondrial dysfunction and secondary nuclear alterations.

The study includes a large number of experiments and the results are well presented

Since the discovered of the first gene (LMNA) linked to Mandibuloacral Dysplasia (MAD) by Novelli and collaborators in 2002, the genetic heterogeneity has been a great challenge for genetic researchers. In the same time, the LMNA gene became the molecular cause of different systemic laminopathies such as Hutchinson Gilford progeria syndrome (HGPS) or atypical progeria syndrome (APS). The introduction of next-generation sequencing technology has led to discover new progeroid genes via expanded gene panels and exome or genome sequencing.

On the basis of complex clinical phenotype of the patients described in the manuscript, the authors suggests to have found a new gene causing a progeroid form of mandibuloacral dysplasia, after MADA and MADB types.

Major comments:

To define the criteria to classify the progeroid phenotype is complex: some features involve the same body systems but the onset, course and severity of the symptoms vary, and other features are unique (Cenni et al, 2018; Hennekam 2006). For this reason:

- in the Results, the author have to mention and discuss (also in supplementary Table 1) the third form of MAD associated to deafness, progeroid features and lipodystrophy, known as MDPL syndrome with an autosomal dominant model of inheritance and caused by POLD1 mutations.
- The cellular phenotype, in particular the mitochondrial defects have to been compared to the known progeroid syndromes but it isn't unclear the comparison only with HGPS cells. After all, the MADaM patients presents overlapping phenotype with MADA, MADB, MDPL and HGPS patients
- MADaM2 and MADaM3 patients are 14y and 2y old. There is an age-related severity of cellular phenotype?

- Do the deformed nuclei accumulate prelamin A?

Minor Remarks

Only coding DNA reference sequence: The manuscript does not mention the reference sequence(s) used for the intronic variant descriptions. Please mention the appropriate Accession Version numbers (examples are: NG_0123456.3, LRG_123....).

Are the identified MTX2 variants reported in gnomAD database? Please check in the variants also in these exome and genome sequencing data.

Please use also Varsome (<https://varsome.com/>) as tool for the MTX2 variant annotations

Please, avoid expressions as "bird-like facies" and use the clinical terms to describe facial features reported in Hennekam R, AJMG Part A 140A:2603–2624 (2006)

In the supplemental figure 3, the authors reported the MTX2 gene structure in exons and introns: the c.544-1G>C is reported in intron 8, while in the manuscript the nucleotide change is attributed to intron 9

Please check the mutations in ZMPSTE24 causing MADB in supplementary Table 1

REVISION OF MANUSCRIPT NCOMMS-19-397788-T

Elouej et al., "Loss of MTX2, responsible for a novel severe progeroid mandibuloacral dysplasia, links mitochondrial dysfunction to altered nuclear morphology."

POINT BY POINT RESPONSES TO THE REFEREES' COMMENTS

First of all we would like to sincerely thank the three Reviewers for their time and thorough analysis of the manuscript.

Their comments helped us to greatly improve the quality of our work, through additional and important workup that we performed in its entirety, excepted for one point (please see below in the point-by-point responses and in the corresponding novel Supplementary figures).

Following the suggestions of the Reviewers, the overall quality of the figures has been improved and the introduction and discussion sections have been deeply reworked, in order to increase the clarity and the critical analysis of the results presented.

Additionally, we would like to inform the Reviewers that during the review process we could identify a 7th patient affected with MADaM syndrome, from Ecuador, carrying a fifth null homozygous mutation. We were thus authorized by the Editor to add the description of the patient and the novel co-authors to the article.

Further functional data from two patients of family MADM4 were also added to the manuscript (novel Supplementary Figure 5).

As well, novel results concerning delayed developmental patterns and reduced reproductive fitness of the *mtx-2* KO *C. elegans* model were added to the work, contributing to prove the noxious effects of the gene's inactivation in this respect.

Reviewer #1 (Remarks to the Author):

In the article entitled "Loss of MTX2, responsible for a novel severe progeroid mandibuloacral dysplasia, links mitochondrial dysfunction to altered nuclear morphology" the authors report the identification of new null mutations in MTX2 leading to a novel severe premature aging syndrome, named MADaM. Furthermore, they have characterised at a cellular level, both structure and metabolism alterations, due the loss of MTX2.

The findings reported in this article are clear and the choice of the performed experiments is in line with the overall purpose of the study. However, there are several concerns that need to be addressed by the authors.

Major:

The introduction looks more like an abstract. We invite the authors to use that part of the article as abstract and write an appropriate introduction.

These suggestions have been followed

We suggest moving lines 70-77 and 106-116 from the results section to introduction with a suitable integration of the missing parts.

These suggestions have been followed

For a better understanding, it would be helpful for the reader to have the results section divided into paragraphs.

This suggestion has been followed

The discussion/conclusion paragraph is missing.

We suggest to use lines 217-235 as discussion paragraph.

Lines 217/219 should be at the end of the discussion as future perspectives.

These suggestions have been followed and the discussion/conclusion section has been developed and improved.

The sentence "Furthermore, MADaM may become a useful model to study frequent age-related disorders..." (lines 235-237), is highly speculative and lacks of appropriate introduction and a deeper discussion. The reader is left wondering about the possible connections between the described mutations with cancer, neurodegenerative and cardiovascular diseases. Please, support your speculations with evidences.

Several references have been added, in order to support the statements.

Why MADaM is correlated with age related disorders? We agree that the referred phenotype "...converge to a systemic premature aging phenotypes" but a deeper explanation will facilitate the reader. Please provide direct evidences in both introduction and discussion.

Arguments were developed both in the introduction and the discussion linking MADaM syndrome with other premature aging syndromes characterized by mandibuloacral dysplasia, both on the clinical and cellular pathophysiological levels. Additionally, patients MADM2 and 3 unfortunately passed away prematurely, respectively in their second and first decades (supplementary data), due to a major cardiovascular burden including increased vascular stiffness, hypertension and peripheral hypoxia causing renal, cardiac and respiratory insufficiencies, recalling age-related organ dysfunction, but in a much more severe fashion.

The absence of MTX2 results into a lack of expression of MTX1, even if the MTX1 is not affected (data not shown). It would be nice to have evidences about the possible transcriptional (qPCR on MTX1 cDNA) or translational (proteasome inhibition and confirmation of MTX1 presence or its ubiquitination) regulation of MTX1 transcript.

qPCR on *MTX1* cDNA has been performed on two patients' cell lines vs controls, showing no transcriptional *MTX1* deregulation (additional Supplementary Figure 6). Conversely, although very interesting, proteasome inhibition could not be performed for this revision but is evoked in the discussion and is foreseen for future work).

Since the authors suggest that MTX2 absence impairs cyto-nucleo-skeletal network (line 181-185), we wonder why actin is used as reference protein for western blot analysis. We suggest using GAPDH or another well-characterised reference protein.

It would be nice, since it is evident by the WB in fig. 2a that actin is differentially expressed by WT and MADM2 and MADM3 (MTX1 panel), to have immunofluorescence images of the cytoskeleton of patients' fibroblasts.

Lastly, we are wondering why for each tested protein the authors show the actin. Is it not part of the same membrane? Do all the proteins have the same MW? Please explain.

We thank Mrs Giorgi for these very legitimate and useful observations and are grouping below the answers to the questions. Indeed, actin was used since it is one of the most widely used housekeeping proteins for WB normalization, it is not part of the same mitochondrial membrane and has a different MW compared to MTX2 and MTX1 (about 42 kDa for actin, 30 kDa for MTX2 and 51.5 kDa for MTX1).

Nonetheless, given the results of the second WB in Fig. 2a, supposedly altered cyto-nucleo-skeletal network interactions and although actin quantities seemed conserved in other western blot studies (other panels of Figure 2a, Figure 2d) among patients and control cell lines, it was of high importance to answer the question asked by the reviewer.

We thus checked by a supplementary and independent study whether actin quantities and distribution were actually conserved in MADaM patients' fibroblasts, allowing us, in the affirmative, to use this protein as an internal normalization protein as we had done in several parts of the article. As you can see in the **novel Supplementary Figure 9, actin levels (evaluated by WB) and distribution (evaluated by IF)** were indeed conserved in patients, making it legitimate to use actin as an internal normalization protein in those patients. We thus suppose that, if there's an altered cyto-nucleo-skeletal network, it involves other proteins and structures; this has also been discussed.

Moreover, lines 181-185, mainly speculative, should be moved in the discussion section (otherwise cyto-nucleo-skeletal interactions alteration should be proved within the article).

This suggestion has been followed

Regarding the WB analysis, we found that the representative images used do not reflect the journal standards, therefore we invite the authors to choose better ones.

The resolution of the figures was improved.

All immunofluorescence images have a too low resolution, the same applies also to all the figures (supplementary fig. 3 is impossible to read). Please provide images with higher dpi (300 to 600).

The resolution of the figures was improved.

The authors provide evidences of severe mitochondrial impairment upon MTX2 absence, it could be interesting and more complete whether the authors could integrate mitochondrial respiration and potential data with calcium measurements.

We thank the reviewer for this comment. Nonetheless, as it can be seen in Fig.3, OXPHOS respiration data and membrane potential measurements have already been obtained on patients' fibroblasts, showing that at least some of the evaluated parameters were impaired : reduced (R-O)/F Ratio in both patients explored, reduced R/F Ratio and increased mitochondrial membrane potential in one patient. Calcium measurements were indeed not performed at this stage but we think that they are not likely to show major variations given that calcium handling defects have been well described in disorders like Wolfram syndrome (e.g. PMID: 30352948), which does not share clinical features with our patients'. So, while preliminary data were obtained in this first work on patients' cells, we plan to perform more extensive morphological and functional studies on mitochondrial respiratory function, including ATP production and calcium measurements that will allow to explore more deeply the patients' mitochondrial phenotype in the future. The fact that those studies are warranted was stated in the discussion.

Minor:

Line 60: Mandibuloacral

Please note that this is the correct spelling of the word.

Line 88: "...by pairing the patients on a phenotypic basis." Please explain.

The sentence was developed in the text in order to make it clearer.

Line 97: "...were found heterozygous for the mutation,...."

The sentence was changed in the text

Line 115-116: ...triggered...

The change was made in the text

Line 120-123: remove by RT-PCR. If it is sequence analysis is better to refer to sequencing.
Please clarify when you talk about cDNA mutation or genomic alterations (same also in supp. Fig. 3)

The requested changes were made both in the main text and in Supp. Fig. 4 (previous Supp. Fig. 3) in order to clarify the studies performed.

Line 127-129: The sentence is intricate. Please rephrase.

The sentence was rephrased.

Line 155: Please remove known.

Done, thank you

Line 197-198: "...confirming that MTX2-deficient...." This sentence is speculative therefore should be moved to the discussion.

Thank you again for the comment, allowing to ameliorate the quality of the manuscript. This speculative sentence and the following one were moved to the discussion.

Line 442: consanguineous.

The spelling was corrected.

Line 454: WB quantification should be done conservatively: please normalize or on REVERT or on reference protein (no actin, see comment above). Otherwise, give a valid explanation.

Please see the explanations noted above.

Figure 2: It is not clear if the statistic described in line 465-468 apply only to panel d or also to panel a. Which is the n for the WB of panel a?

the statistic described in lines 465-468 applies to panels c and d and this has been mentioned in the novel legend. The number of WBs for each immunoblotted protein has been detailed in the legend.

Line 480: "...Student's t test was performed comparing patients' value to control."

The change was made.

Figure 4: Please number the panels from left to right, top to bottom.

The changes were made.

Figure 5n: The figure selected are not clear. Please provide better ones.

The changes were made.

Supplementary information Line 46: progressive progressive

The change was made.

Supplementary information Line 268: Which counting methods was used for trypan blue exclusion test?

The Countess™ Automated Cell Counter was used to count the number of live and dead cells after trypan blue staining and the method was added in the materials and methods section.

Supplementary fig. 3a: Alignment with C. elegans sequence is missing. Xenopus l.

The alignment with C. elegans amino acid sequence was added, together with that of Xenopus tropicalis (instead of laevis).

Supplementary fig. 3b: RT-PCR refers to reverse transcription and not to PCR amplification. Please refer to PCR in the legend. What does T stand for?

The changes were made and we explained what T stands for.

Supplementary fig. 3c: The introns numbering is not clear: c.208...should be on intron 4 and c.544 on intron 9. Please name the mutations in the same way.

Supplementary fig. 3c is correct but there was indeed an error in the text concerning the localization of mutation c.544-1G>C: this is located in intron 8 and the error was corrected. The c.208... mutation is located in intron 4 (both in the figure and in the text).

Supplementary fig. 4: Please use better images.

The quality of the images was improved in the novel Supplementary fig. 7

Supplementary fig. 5: Please use a better resolution.

The quality of the images was improved in the novel Supplementary fig. 10 (to obtain better figures, the resolution was set at 600 dpi using Adobe Photoshop CS4).

Carlotta Giorgi

Reviewer #2 (Remarks to the Author):

The major claims of the paper submitted by the corresponding author Annachiara De Sandre-Giovannoli are that a form of Mandibuloacral Dysplasia is associated with homozygous null mutations in a mitochondrial membrane gene, MTX-2, encoding for the outer mitochondrial membrane protein metaxin-2 and the pathogenetic effect of the mutation is linked altered morphofunctional organization of mitochondrial network, loss of cellular response to apoptotic stimuli, activation of an unconventional autophagic pathway and nuclear morphological defects. This is the first paper linking Mandibuloacral dysplasia to mitochondrion dysfunction, although impaired mitochondrial activity has been observed in the most severe progeroid laminopathy, Hutchinson-Gilford progeria and in a familial partial lipodystrophy associated with LMNA mutations, the FPLD2. The results are mostly unexpected and novel and, since Mandibuloacral dysplasia is a progeroid syndrome with partial lipodystrophy and bone resorption, they will be of interest to researchers in the field of progeroid syndromes and in the wider field of normal and pathological ageing research, research on adipose tissue dysfunction and bone turnover disorders.

A few interesting points need to be addressed:

- The authors nicely show that nuclear morphology is altered, while lamin A/C is not affected. Based on published data, which need to be cited (Filesi et al 2005; Lombardi et al., 2007; Camozzi et al 2012), prelamin A accumulation could be involved. Confocal microscopy analysis of prelamin A immunostaining with diverse antibodies is needed to rule out or acknowledge this hypothesis.

We thank the reviewer for this comment. Prelamin A western blot and indirect immunofluorescence studies were performed on MADaM patients vs WT and MAD-B patient fibroblast cell lines. These additional experiments clearly show that prelamin A is present in MAD-B fibroblasts but not in any of the other fibroblasts (novel Supplementary Figure 8).

- An important paper shows that mitochondrial dysfunction interferes with chromatin epigenetic modifications and directs longevity-associated mechanisms (Tian et al, Mitochondrial Stress Induces

Chromatin Reorganization to Promote Longevity and UPRmt, Cell 2016). This paper must be discussed and chromatin marks as H3K9-dimethyl need to be analyzed, also considering that H3K9 epigenetic defects have been described in Mandibuloacral dysplasia (Filesi et al., 2005; Camozzi et al., 2012).

Given the published data mentioned by the reviewer and other articles showing that also H3K9-trimethylation (H3K9-3me) levels are low in HGPS (Shumaker et al, 2006) and in progeroid laminopathies as MADA (Filesi et al. 2005), those were evaluated by western blot in MADaM fibroblast cell lines vs WT and HGPS fibroblasts (novel Supplementary Fig. 10). These studies confirmed, as expected, that H3K9-3me levels were reduced in HGPS fibroblasts, while they were conserved in MADaM patients' fibroblasts.

Reviewer #3 (Remarks to the Author):

Confirming the genetic heterogeneity linked to progeroid syndromes, the authors reported four homozygous null mutations in MTX2 gene in six patients presenting a novel severe premature aging syndrome characterized by growth retardation, bone resorption, arterial calcification, renal glomerulosclerosis and severe hypertension. This new progeroid gene encodes for an outer mitochondrial membrane protein and in vitro functional studies demonstrated that loss of MTX2 leads to mitochondrial dysfunction and secondary nuclear alterations.

The study includes a large number of experiments and the results are well presented. Since the discovery of the first gene (LMNA) linked to Mandibuloacral Dysplasia (MAD) by Novelli and collaborators in 2002, the genetic heterogeneity has been a great challenge for genetic researchers. In the same time, the LMNA gene became the molecular cause of different systemic laminopathies such as Hutchinson Gilford progeria syndrome (HGPS) or atypical progeria syndrome (APS). The introduction of next-generation sequencing technology has led to discover new progeroid genes via expanded gene panels and exome or genome sequencing. On the basis of complex clinical phenotype of the patients described in the manuscript, the authors suggest to have found a new gene causing a progeroid form of mandibuloacral dysplasia, after MADA and MADB types.

Major comments:

To define the criteria to classify the progeroid phenotype is complex: some features involve the same body systems but the onset, course and severity of the symptoms vary, and other features are unique (Cenni et al, 2018; Hennekam 2006). For this reason:

-in the Results, the authors have to mention and discuss (also in supplementary Table 1) the third form of MAD associated to deafness, progeroid features and lipodystrophy, known as MDPL syndrome with an autosomal dominant model of inheritance and caused by POLD1 mutations.

We thank the reviewer for this comment. Indeed, the description of MDPL was missing and the disease was thus mentioned and described both in the introduction and in the Supplementary Table 1.

- The cellular phenotype, in particular the mitochondrial defects have to be compared to the known progeroid syndromes but it isn't unclear the comparison only with HGPS cells. After all, the MADaM patients presents overlapping phenotype with MADA, MADB, MDPL and HGPS patients

This comment is very legitimate but, in this initial exploratory work we focused on HGPS fibroblast abnormalities as a comparison to those of our patients because, among the progeroid laminopathies, HGPS is the best characterized at the cellular and molecular levels, allowing easier first-phase comparisons of pathophysiological elements. Of course we agree that larger studies extending to the other MAD syndromes are warranted.

-MADaM2 and MADaM3 patients are 14y and 2y old. There is an age-related severity of cellular phenotype?

No, this was not observed in our study. In fact, patient MADM3, although younger, presented with an even more severe clinical phenotype than patient MADM2 and this may be one of the reasons leading to the absence of major difference among the cellular phenotypes of the patients.

- Do the deformed nuclei accumulate prelamin A?

Please see the answer to the same question from Reviewer #2 and the novel Supplementary Figure 8.

Minor Remarks

Only coding DNA reference sequence: The manuscript does not mention the reference sequence(s) used for the intronic variant descriptions. Please mention the appropriate Accession Version numbers (examples are: NG_0123456.3, LRG_123....).

Thank you for this comment. Further appropriate Accession numbers were provided in the Sanger sequencing confirmation paragraph, in the Supplementary Material file.

Are the identified MTX2 variants reported in gnomAD database? Please check in the variants also in these exome and genome sequencing data.

None of them are reported in gnomAD and this was stated in the manuscript text.

Please use also Varsome (<https://varsome.com/>) as tool for the *MTX2* variant annotations.

Thank you, this was done and the results were given in the manuscript text.

Please, avoid expressions as “bird-like facies” and use the clinical terms to describe facial features reported in HennekamR, AJMG Part A 140A:2603–2624 (2006)

Thank you for the correction. The terms were replaced with appropriate ones in the clinical supplementary table 1, as well as in the main text and supplementary material.

In the supplemental figure 3, the authors reported the *MTX2* gene structure in exons and introns: the c.544-1G>C is reported in intron 8, while in the manuscript the nucleotide change is attributed to intron 9.

The mutation c.544-1G>C is indeed located in intron 8 and the error was corrected in the manuscript.

Please check the mutations in *ZMPSTE24* causing MADB in supplementary Table 1

Thank you, those were checked and corrected: the homozygous c.281T>C, p.(Leu94Pro) from Ben Yaou et al., 2011 was inserted instead as an example.

REVIEWERS' COMMENTS second round:

Reviewer #1 (Remarks to the Author):

The authors addressed satisfactorily almost all the points I raised.

Just few minor points need to be further revised

Figure 2: The blot chosen are the same ones of the previous version. We kindly ask to provide better blots according to the journal standards (if is not possible to repeat the experiments please provide an explanation). In the figure legend 2a there is no need to specify for the normalization used since there is no quantification.

Figure 5n: The authors stated that the figure was changed but it was not. It is not clear the difference between tubular and fragmented (maybe due to overexposure of the image). Please provide an explanation for this.

Reviewer #2 (Remarks to the Author):

The major claims of the paper are that mutations in a mitochondrial membrane protein cause a progeroid syndrome and, at the cellular level, affect nuclear morphology. These findings are novel and will they be of interest to researchers in the field of normal and pathological ageing.

The paper also opens new biological questions, regarding how mitochondrial defects may have an impact on nuclear phenotype.

Reviewer #3 (Remarks to the Author):

The authors responded adequately to what was requested. Therefore this Reviewer has nothing to ask further

Point-by-point response to referees

Reviewer #1 (Remarks to the Author):

The authors addressed satisfactorily almost all the points I raised.

Just few minor points need to be further revised

Figure 2: The blot chosen are the same ones of the previous version. We kindly ask to provide better blots according to the journal standards (if is not possible to repeat the experiments please provide an explanation).

Indeed those western blots could not be reproduced during the time allowed for revision of the manuscript, together with the other experiments requested, also because the cell lines of MADaM patients grow very slowly.

On the other hand, even if not too “beautiful”, those blots can be trusted; indeed, in the revised version of the paper we provided evidence that using actin for normalization is reliable, given that overall actin levels are not affected in patients' fibroblasts; finally, supplementary western blots were provided in the source data files.

In the figure legend 2a there is no need to specify for the normalization used since there is no quantification.

Thank you, that has been corrected.

Figure 5n: The authors stated that the figure was changed but it was not. It is not clear the difference between tubular and fragmented (maybe due to overexposure of the image). Please provide an explanation for this.

The figure was changed in that it was provided at a higher resolution than the previous version.

Nonetheless, in order to make the differences among the mitochondrial networks clearer, we improved the figure by zooming into the networks and by reducing the excessive exposure, similarly in all panels.

Reviewer #2 (Remarks to the Author):

The major claims of the paper are that mutations in a mitochondrial membrane protein cause a progeroid syndrome and, at the cellular level, affect nuclear morphology. These findings are novel and will they be of interest to researchers in the field of normal and pathological ageing.

The paper also opens new biological questions, regarding how mitochondrial defects may have an impact on nuclear phenotype.

Reviewer #3 (Remarks to the Author):

The authors responded adequately to what was requested. Therefore this Reviewer has nothing to ask further